# Transcriptional network involving ERG and AR orchestrates Distal-less homeobox-1 mediated prostate cancer progression

Sakshi Goel [1], Vipul Bhatia [1], Sushmita Kundu[1], Tanay Biswas[1], Shannon Carskadon[2], Nilesh Gupta[3], Mohammad Asim[4], Colm Morrissey[5], Nallasivam Palanisamy[2] & Bushra Ateeq [1,6✉]

Distal-less homeobox-1 (DLX1) is a well-established non-invasive biomarker for prostate cancer (PCa) diagnosis, however, its mechanistic underpinnings in disease pathobiology are not known. Here, we reveal the oncogenic role of DLX1 and show that abrogating its function leads to reduced tumorigenesis and metastases. We observed that ~60% of advanced-stage and metastatic patients display higher *DLX1* levels. Moreover, ~96% of *TMPRSS2-ERG* fusion-positive and ~70% of androgen receptor (AR)-positive patients show elevated *DLX1*, associated with aggressive disease and poor survival. Mechanistically, ERG coordinates with enhancer-bound AR and FOXA1 to drive transcriptional upregulation of *DLX1* in ERG-positive background. However, in ERG-negative context, AR/AR-V7 and FOXA1 suffice to upregulate *DLX1*. Notably, inhibiting ERG/AR-mediated *DLX1* transcription using BET inhibitor (BETi) or/ and anti-androgen drugs reduce its expression and downstream oncogenic effects. Conclusively, this study establishes *DLX1* as a direct-target of ERG/AR with an oncogenic role and demonstrates the clinical significance of BETi and anti-androgens for DLX1-positive patients.

[1] Molecular Oncology Laboratory, Department of Biological Sciences and Bioengineering, Indian Institute of Technology Kanpur, Kanpur, U.P., India. [2] Vattikuti Urology Institute, Department of Urology, Henry Ford Health System, Detroit, MI, USA. [3] Department of Pathology, Henry Ford Health System, Detroit, MI, USA. [4] Department of Clinical and Experimental Medicine, Faculty of Health and Medical Sciences, University of Surrey, Guildford, UK. [5] Department of Urology, University of Washington, Seattle, WA, USA. [6] The Mehta Family Center for Engineering in Medicine, Indian Institute of Technology Kanpur, Kanpur, U.P., India. ✉email: bushra@iitk.ac.in

Recurrent gene rearrangements involving the androgen-regulated transmembrane protease serine 2 (*TMPRSS2)* and ETS transcription factor, v-ets erythroblastosis virus E26 oncogene homolog (*ERG*) occur in ~50% of prostate cancer (PCa) patients[1]. Aberrant overexpression of ERG controls a transcriptional network linked to PCa development[2,3], increased metastatic potential, and associate with poor clinical outcome[4]. In *TMPRSS2-ERG* fusion-positive PCa, ERG recruitment onto the *SOX9* promoter opens up cryptic androgen receptor (AR) binding sites on the *SOX9* enhancer thereby regulating its expression, while the loss of SOX9 results in reduced ERG-mediated oncogenicity[5]. Nonetheless, other critical partners that govern ERG-mediated oncogenesis remain largely unexplored.

Considering the critical role of AR signaling in the development and progression of PCa, inhibitors targeting androgen synthesis (abiraterone acetate) and AR antagonists (bicalutamide, enzalutamide) are used in the first line of treatment termed androgen deprivation therapy (ADT), often administered to advanced stage PCa patients[6]. Although, prolonged administration of ADT results in an inevitable cancer relapse due to the selection pressure of drugs, eventually progressing to aggressive castration-resistant prostate cancer (CRPC) stage[6]. Recent drugs such as apalutamide and darolutamide prolong the metastatic-free survival of CRPC patients[7]. In addition, the Food and Drug Administration (FDA) has also approved the use of apalutamide for non-metastatic, castration-sensitive patients[8]. Mounting evidence suggests sustained AR activity in CRPC, owing to numerous mechanisms including *AR* amplification, gene mutations, intra-tumoral androgen synthesis, and expression of constitutively active *AR* splice variants (AR-Vs)[6]. Moreover, patients treated with enzalutamide and abiraterone showed increased levels of AR-V7 in the circulating tumor cells, and shorter time to biochemical relapse compared to AR-V7 negative cases[9]. Unlike full-length AR (AR-FL), the splice variant AR-V7 functions in a ligand-independent manner and remains constitutively active driving androgen-independent growth and disease progression[10]. Interestingly, a recent study showed a distinct AR-V7 cistrome that governs cell-context-dependent gene expression[11]. Recent studies revealed the clinical significance of bromodomain and extra-terminal (BET) domain proteins, which are known transcriptional coactivators of tumor-promoting genes such as AR and are considered as a potential therapeutic target for CRPC treatment[12,13].

The *Distal-less homeobox (DLX)* genes belong to the homeobox-containing family of transcription factors (TFs), which are structural homologs of *Drosophila Distal-less (Dll)*. *DLX1* being a member of the *DLX* family plays an essential role in the development of craniofacial features, jaw, and GABAergic (gamma-aminobutyric acid) interneuron[14]. Deregulation of homeobox genes has been linked to several human malignancies including prostate[15]. In hematopoietic cells, DLX1 impedes downstream TGF-β-mediated signaling pathways by interacting with Smad4[16]. In PCa, DLX1 is known to functionally interact with β-catenin and regulate downstream β-catenin/TCF4 signaling pathway[17]. Furthermore, *DLX1* has been validated as a PCa biomarker across clinically independent cancer cohorts, wherein *DLX1* and *HOXC6* accurately predict high-grade disease[18]. However, the underlying regulatory mechanism that drives DLX1 upregulation and its functional role in PCa progression remain poorly understood.

Here, we uncover the molecular mechanisms underlying the upregulation of DLX1 in PCa and reveal oncogenic functions associated with it. Our data identify DLX1-mediated downstream biological processes that operate in PCa tumorigenesis and show its oncogenic role in disease progression and metastasis. We also demonstrate the role of ERG, AR, and FOXA1 as key transcriptional regulators involved in *DLX1* overexpression in PCa. Lastly,

we show that disrupting ERG/AR transcriptional circuitry with BET inhibitor (BETi) or in combination with anti-androgen could attenuate DLX1-mediated tumorigenesis. Collectively, this study highlights the importance of DLX1 as an effective therapeutic target in PCa patients irrespective of *TMPRSS2-ERG* fusion status.

## Results

**DLX1 imparts oncogenic properties and promotes PCa progression**. To study the association between increased DLX1 levels and PCa progression, we analyzed RNA-sequencing (RNA-Seq) data using publicly available clinical genomics data repository viz. The Cancer Genome Atlas Prostate Adenocarcinoma (TCGA-PRAD) dataset. Interestingly, higher expression of *DLX1* was observed in patients with primary PCa compared to matched normal tissue (Fig. 1a). Similarly in other clinical genomics datasets (GSE35988[19] and GSE80609[20]) an increased expression of *DLX1* transcript was observed in advanced stage aggressive cancers compared to benign (Fig. 1b and Supplementary Fig. 1a). In agreement with this, patients with higher *DLX1* expression (*DLX1*[Hi]) experienced poor survival probability compared to those with lower *DLX1* expression (*DLX1*[Lo]) (Fig. 1c). Likewise, elevated *DLX1* levels were observed in PCa cell lines (22RV1, VCaP, and PC3) that represent CRPC compared to LNCaP cells, an androgen-responsive, and RWPE1, a benign and immortalized prostate epithelial cell line (Supplementary Fig. 1b). Also, the read coverage of *DLX1* transcript from the RNA-Seq data of RWPE1[21], 22RV1[22], and VCaP[22] cell lines showed similar trends (Fig. 1d). To understand the functional significance of DLX1 in PCa, we ectopically overexpressed DLX1 in RWPE1 cells and confirmed its overexpression both at transcript and protein levels (Supplementary Fig. 1c). Interestingly, a significant increase in cell proliferation in RWPE1-DLX1 cells compared to control was observed (Fig. 1e). Similarly, DLX1 overexpression markedly increased foci forming ability and migratory properties of RWPE1 cells (Fig. 1f, g). In contrast, to understand the functional role of DLX1 using a loss-of-function model, CRISPR/Cas9 mediated gene knockout of *DLX1* was performed in 22RV1 cells (Supplementary Fig. 1d), and *DLX1* knockout (KO) was confirmed by genomic amplification of the CRISPR/Cas9 target gene sequence (Supplementary Fig. 1e). Three independent *DLX1*-KO clones, namely 22RV1-*DLX1*-KO (C-1, C-2, and C-3) showing loss of DLX1 expression both at transcript and protein levels (Supplementary Fig. 1f) were selected to determine any phenotypic changes in these genetically engineered lines. Notably, a marked reduction in proliferation rates of 22RV1-*DLX1*-KO cells was observed compared to control (Fig. 1h). Similarly, a significant reduction (~60%) in the cell migratory and foci forming properties was observed in 22RV1-*DLX1*-KO cells (Fig. 1i and Supplementary Fig. 1g). We also observed a ~4–5-fold decrease in the anchorage-independent growth of 22RV1-*DLX1*-KO cells compared to control (Fig. 1j).

To delineate the biological pathways orchestrated by DLX1 in PCa, we performed a microarray-based global gene expression profiling of 22RV1-*DLX1*-KO and 22RV1-SCR cells. Database for Annotation, Visualization and Integrated Discovery (DAVID, https://david.ncifcrf.gov/) bioinformatics analysis on the differentially expressed genes (log₂ fold-change >0.6 or <−0.6, 90% confidence interval) revealed significantly up- and downregulated biological processes engaged by DLX1 (Supplementary Data 1). As anticipated, genes involved in proliferation and migration of cells, and stem-cell population maintenance were negatively enriched in 22RV1-*DLX1*-KO cells, while genes associated with cell cycle regulation and antigen processing and presentation were upregulated (Fig. 1k). Subsequent gene set enrichment analysis

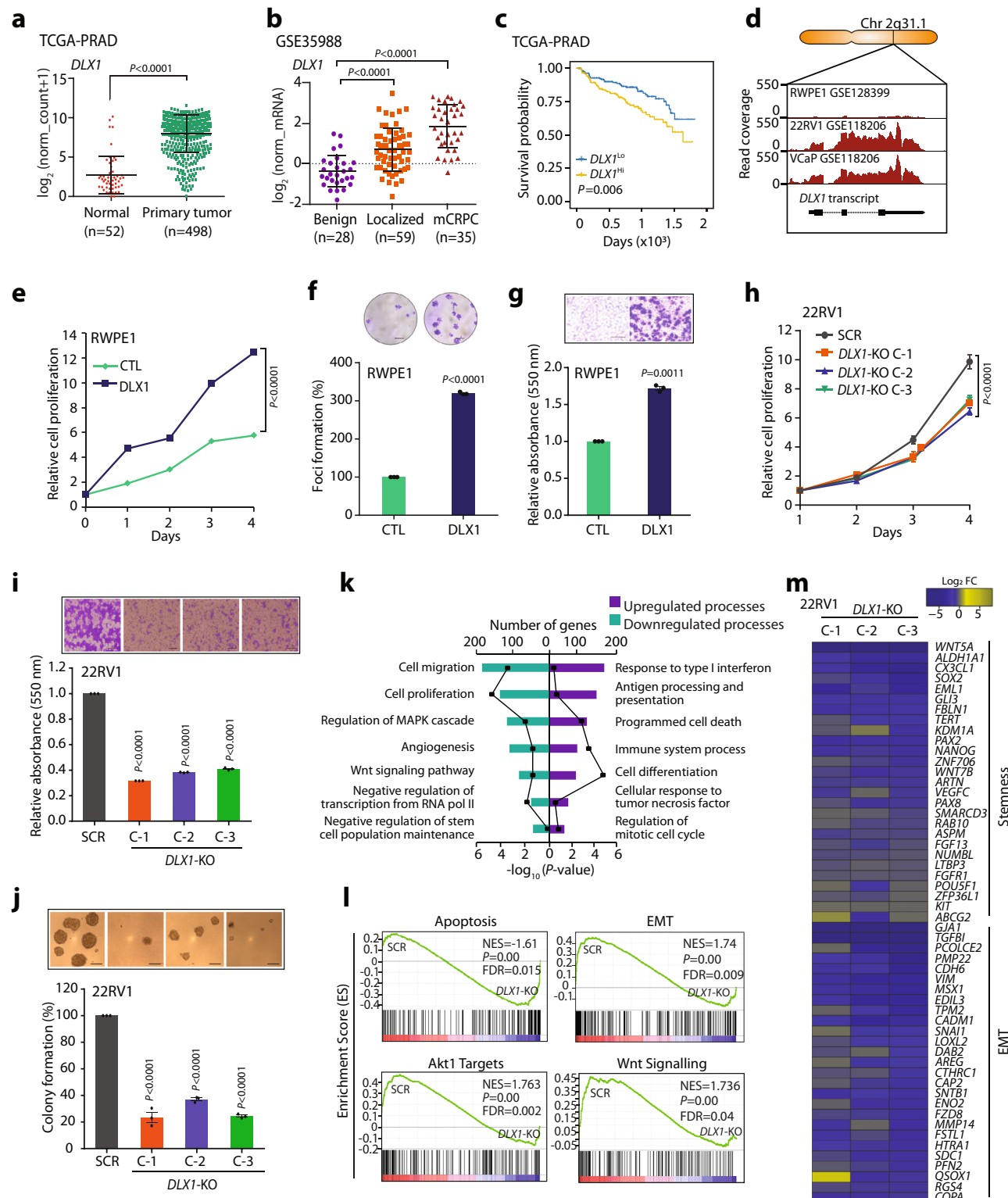

(GSEA) showed enrichment of genes involved in the apoptotic pathway in *DLX1*-KO cells, whereas gene sets involved in epithelial-to-mesenchymal transition (EMT), cancer stemness, active Akt, and Wnt signaling were enriched in the control cells (Fig. 1l, m). To further explore the oncogenic role of DLX1 in aggressive PCa, we analyzed RNA-Seq data (GSE78913)[23] for the transient knockdown of *DLX1* in LNCaP derivative osteotropic cell line C4-2B, and as speculated *DLX1* was found to be one of the top five downregulated genes as depicted in the volcano plot

(Supplementary Fig. 1h). In addition, GSEA and pathway analysis of the differentially expressed genes showed deregulation of important pathways associated with cancer, such as upregulation of apoptosis and cell cycle arrest (Supplementary Fig. 1i, j). Further, functional characterization of differentially expressed genes in the 22RV1-*DLX1*-KO cells by an overlapping network-based analysis revealed significant ($P < 0.005$, FDR $< 0.1$) down-regulation of EMT, cell cycle and DNA damage pathways (Supplementary Fig. 2). Taken together, our findings implicate

**Fig. 1 High DLX1 expression associates with poor prostate cancer prognosis and promotes disease progression. a** Dot plot showing *DLX1* expression in PCa patients ($n = 498$) and matched normal ($n = 52$) in TCGA-PRAD RNA-Seq dataset, data represents log2 (norm_count+1), center depicts mean ± (standard deviation) SD ($P < 0.0001$). **b** Dot plot of *DLX1* expression using microarray profiling data (GSE35988) comprising benign ($n = 28$), localized ($n = 59$), and mCRPC ($n = 35$) patient specimens, data represents log2 (norm_mRNA), center depicts mean ± SD ($P < 0.0001$). **c** Kaplan–Meier plot showing survival probability in TCGA-PRAD ($n = 498$) dataset categorized in high *DLX1* (*DLX1*Hi) and low *DLX1* (*DLX1*Lo) expression. **d** RNA-Seq data showing *DLX1* transcript read counts in publicly available datasets (GSE128399 and GSE118206). **e** Cell proliferation assay using isogenic RWPE1 cells overexpressing DLX1 at indicated time-points ($P < 0.0001$). **f** Foci formation assay using same cells as **e** ($P < 0.0001$). **g** Boyden Chamber Matrigel migration assay using same cells as **e** ($P < 0.0001$). Representative images for panels **f** (scale bar 500 μm) and **g** (scale bar 100 μm) are shown as inset. **h** Cell proliferation assay using 22RV1-*DLX1*-KO (C-1, C-2, and C-3 are independent clones) and control cells at indicated time-points ($P < 0.0001$). **i** Boyden Chamber Matrigel migration assay using same cells as **h** ($P < 0.0001$). **j** Anchorage-independent soft agar assay using same cells as **h** ($P < 0.0001$). Representative images for panels **i** and **j** are shown as inset (scale bar 100 μm). **k** DAVID analysis showing upregulated (right) and downregulated (left) biological processes in 22RV1-*DLX1*-KO against control cells. Bars represent the $-\log_{10}$ (*P*-value) and the frequency polygon (black line) denotes number of genes. **l** Same as **k**, except gene set enrichment analysis (GSEA) plots representing deregulated pathways. **m** Heatmap displaying downregulated genes involved in cancer stemness and EMT in 22RV1-*DLX1*-KO cells compared to control. Data shown from three biologically independent samples ($n = 3$). Data represent mean ± SEM unless specified. For panels, **a** Unpaired Student's two-tailed *t*-test was applied; **b** One-way ANOVA with Dunnett's multiple comparison test was applied; **e** Two-way ANOVA Sidak's multiple comparison test; **f**, **g** Unpaired two-tailed Welch's *t*-test; **h** Two-way ANOVA, Dunnett's multiple comparisons test; **i**, **j** One-way ANOVA, Dunnett's multiple comparisons test was applied. Source data are provided as a Source Data file.

the critical role of DLX1 in imparting oncogenic properties to prostate cells.

**DLX1 elicits biological processes involved in cancer progression.** To investigate the underlying mechanism involved in oncogenic properties imparted by DLX1, we examined the expression of genes involved in EMT at the transcript and protein level. Reduced expression of two archetypal markers of the mesenchymal phenotype, namely Vimentin and Snail, with increased E-cadherin expression, an epithelial marker in *DLX1*-KO cells was observed (Fig. 2a, b). Conversely, increased vimentin expression was observed in DLX1 overexpressing RWPE1 cells (Supplementary Fig. 3a). Consistent with the GSEA data, an increased apoptosis marked by Caspase 3 activation and cleaved poly-(ADP-ribose) polymerase (PARP) in *DLX1*-KO cells along with decrease in the levels of anti-apoptotic Bcl-xL was observed (Fig. 2c). Concurrently, increased expression of late apoptotic markers was noted as shown by a rise in the number of Annexin-V and 7-AAD (7-amino-actinomycin D) stained 22RV1-*DLX1*-KO cells (Fig. 2d). In line with this, *DLX1* silencing in VCaP cells also augment apoptosis as shown by an increase in the number of Annexin-V and 7-AAD positive cells (Fig. 2d and Supplementary Fig. 3b). Next, our flow-cytometry-based cell cycle analysis revealed an increase in the percentage of cells arrested in the S-phase with a concomitant decrease in the G2/M phase cells in *DLX1*-KO compared to control (Fig. 2e). Similarly, small interfering RNA (siRNA)-mediated transient knockdown of *DLX1* in VCaP cells resulted in cell cycle arrest (Fig. 2e). Although, the cell cycle arrest in transiently *DLX1*-silenced VCaP cells was not as robust as observed in CRISPR-based *DLX1* knockout in 22RV1. Furthermore, an increase in the phospho-Akt (pAkt) levels was observed in RWPE1-DLX1 cells (Supplementary Fig. 3a), indicating the critical role of DLX1 in regulating cancer cell survival and proliferation.

In agreement with the microarray data, *DLX1*-KO cells exhibit decreased expression of established stem cell markers such as *POU5F1* (Oct-4), *ABCG2*, *CD117* (c-KIT), and *SOX2* (Fig. 2f). Next, we examined the cell surface expression of two well-known stem cell markers, ABCG2 and CD44 in 22RV1-*DLX1*-KO, *DLX1*-silenced VCaP, and RWPE1-DLX1 cells. Interestingly, a marked reduction in the expression of ABCG2 (~30–60%) and CD44 (~50–80%) was observed in 22RV1-*DLX1*-KO and *DLX1*-silenced VCaP cells, while a robust increase in these markers was noted in RWPE1-DLX1 cells (Fig. 2g). Since increased aldehyde dehydrogenase 1A1 (ALDH1A1) activity is often associated with

cancer stem cell phenotype[24], we performed ALDH activity assay using RWPE1-DLX1 cells that showed a significant increase in ALDH activity (Supplementary Fig. 3c). Conversely, 22RV1-*DLX1*-KO and *DLX1*-silenced VCaP cells showed reduction in the ALDH activity (Fig. 2h, i), indicating a plausible function of DLX1 in promoting cancer stemness. Furthermore, to confirm the oncogenic role of DLX1 in enzalutamide-resistant PCa, we silenced *DLX1* in LNCaP derivative enzalutamide-resistant 42D cells (42D ENZᴿ) (Supplementary Fig. 3d). Interestingly, *DLX1* silencing in 42D ENZᴿ cells resulted in S-phase cell cycle arrest and increased expression of apoptotic markers (Supplementary Fig. 3e, f). Furthermore, *DLX1* silencing was marked with reduced ALDH activity in 42D ENZᴿ cells (Supplementary Fig. 3g). Together, these data highlight the possible role of DLX1 in promoting cancer stemness, EMT, and proliferation in PCa.

To explore the DLX1 target genes which are involved in tumorigenesis, we analyzed publicly available chromatin immunoprecipitation sequencing (ChIP-Seq) data for DLX1 in colorectal cancer LoVo cells[25]. By intersecting the genes associated with DLX1 MACS (Model-based Analysis of ChIP-Seq) peaks and the genes downregulated in 22RV1-*DLX1*-KO cells, we observed 677 common genes including *ALDH1A1*, *HNF1A*, *GATA2*, *CDK4*, *MYC*, *WNT5A* as the putative transcriptional targets of DLX1 (Supplementary Fig. 3h). Moreover, DAVID analysis of the common genes revealed downregulation of important biological pathways associated with cancer progression (Supplementary Fig. 3i). Next, we investigated the expression of PCa associated genes, namely *ALDH1A1*[24], *HNF1A*[26], and *GATA2*[27] in 22RV1-*DLX1*-KO cells, and a significant decrease in their expression compared to control cells was observed (Fig. 2j). A direct role of DLX1 in upregulating the expression of these genes was established by performing ChIP-qPCR, which confirmed the occupancy of DLX1 on the promoters of *ALDH1A1* and *HNF1A*, while reduced DLX1 enrichment was noted in 22RV1-*DLX1*-KO cells (Fig. 2k–m). Further, decrease in the histone H3 lysine 9 acetylation (H3K9Ac), a transcriptional activation mark at the same sites confirmed DLX1-mediated positive transcriptional regulation (Fig. 2l, m). Hence, we establish *ALDH1A1* and *HNF1A* as the direct transcriptional targets of DLX1, which are also known to play critical role in stemness, embryonic development, and carcinogenesis[28,29].

**DLX1 plays pivotal role in tumor growth and metastasis.** To examine the role of DLX1 in tumorigenesis, we performed mice xenograft experiment by implanting 22RV1-*DLX1*-KO or

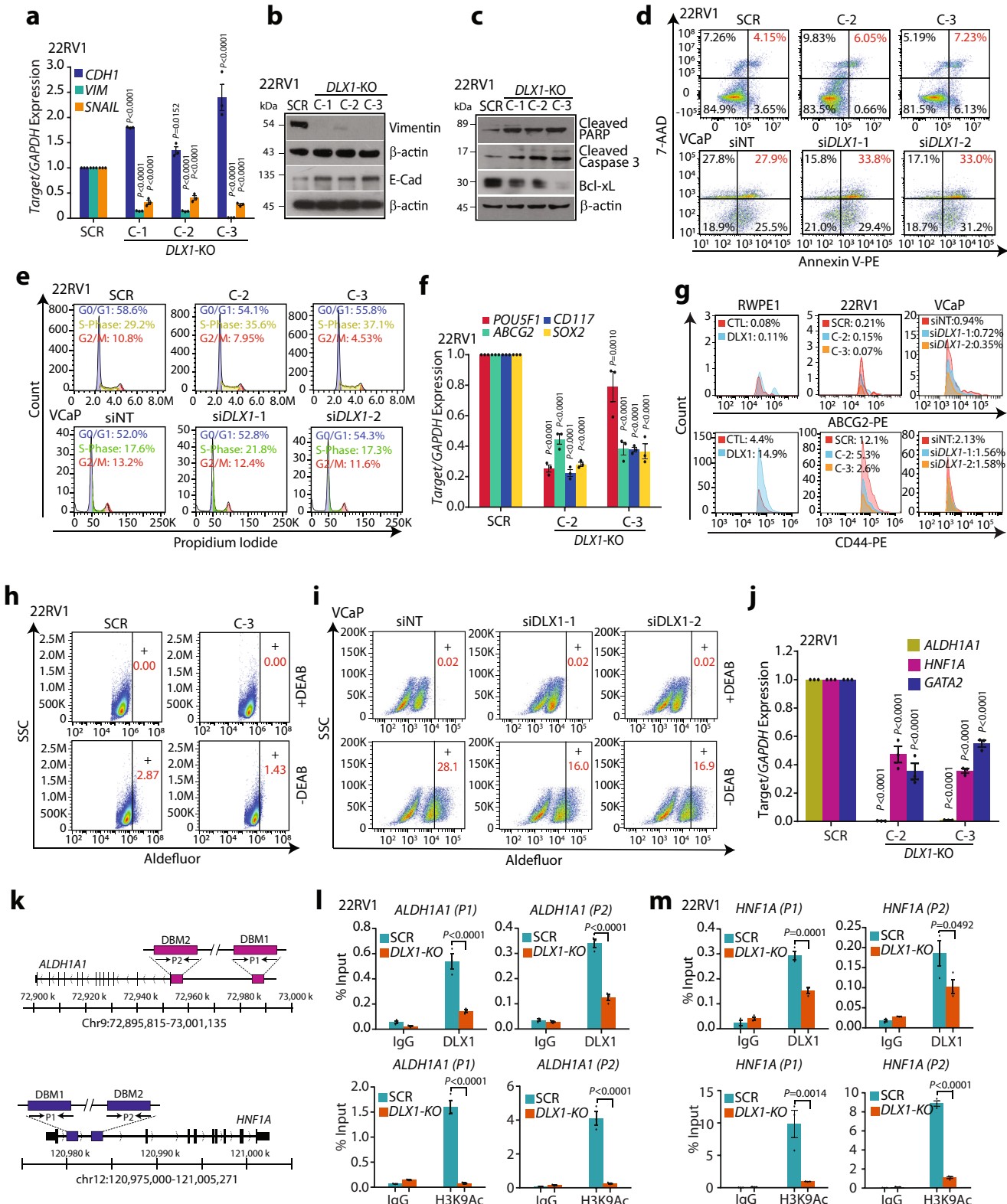

22RV1-SCR control cells subcutaneously in the flank region of immunodeficient NOD/SCID mice and monitored the animals for tumor growth. A significant decrease in tumor growth with a notable ~80% reduction in tumor burden at the end of the study was observed in the 22RV1-*DLX1*-KO cells implanted group compared to the control group (Fig. 3a, b). To examine spontaneous metastasis, lungs and bone marrow were excised from the xenografted mice and were screened for the presence of

human-specific *Alu*-sequence[30,31]. A significant reduction in the number of cells metastasized to the bone marrow and lungs was observed in the 22RV1-*DLX1*-KO cells implanted group (Fig. 3c). We next performed immunohistochemistry (IHC) staining for proliferation (Ki-67) and EMT (Vimentin and E-cadherin) markers on the formalin-fixed paraffin-embedded tumor tissues. A significant reduction in Ki-67 expression, accompanied by reduced Vimentin and increased E-cadherin expression was

**Fig. 2 Genetic ablation of *DLX1* inhibits oncogenic properties. a** Q-PCR data showing expression of EMT markers in 22RV1-*DLX1*-KO and control SCR cells. **b** Immunoblots showing vimentin and E-cadherin using same cells as **a**. β-actin was used as loading control. **c** Same as **b** except for cleaved PARP, cleaved Caspase-3 and Bcl-xL. **d** Flow cytometry-based apoptosis assay using 22RV1-*DLX1*-KO and control cells (top panel) and *DLX1*-silenced VCaP cells (bottom panel). **e** Flow cytometry data for cell cycle distribution using same cells as in **d**. **f** Q-PCR data for stem cell markers using same cells as **d**. **g** Flow cytometry data depicting ABCG2 (top panel) and CD44 (bottom panel) expression in DLX1 overexpressing RWPE1 cells, 22RV1-*DLX1*-KO, and *DLX1*-silenced VCaP cells. **h** Fluorescence intensity of catalyzed ALDH substrate in 22RV1-*DLX1*-KO and control cells. Marked windows show ALDH1 + percent cell population. **i** Same as **h**, except for *DLX1* silenced VCaP cells. **j** Q-PCR data showing expression of target genes in 22RV1-*DLX1*-KO and control cells($P < 0.0001$). **k** Schema showing the chromosomal location of *DLX1* binding motif (DBM1/2) at the *ALDLH1A1* (top) and *HNF1A* (bottom) promoters. **l** ChIP-qPCR data of DLX1 (top panel) and H3K9Ac (bottom panel) on *ALDH1A1* in 22RV1-*DLX1*-KO and SCR control cells ($P < 0.0001$). **m** Same as in **l** except for the *HNF1A* promoter. Data shown from three biologically independent samples ($n = 3$). Data represent mean ± SEM. For panels, **a**, **f**, and **j** Two-way ANOVA, Dunnett's multiple comparison test; **l**, **m** Two-way ANOVA Turkey's multiple comparison test was applied. Source data are provided as a Source Data file.

observed in the *DLX1*-KO group compared to control. Moreover, a significant decrease in the expression of DLX1 target, ALDH1A1 was also recorded in *DLX1*-KO tumor tissues (Fig. 3d, e). Notably, immunostaining for ALDH1A1 in the xenograft tumor tissue was heterogenous, indicating random niches of tumor cells having stem cell-like phenotype.

The role of *DLX* homeobox genes including *DLX1* has been established in skeletal development and bone tumors[32,33]. Moreover, DLX1 has been reported to play a key role in osteoclastogenesis and bone-resorption[34], which contributes to osteolytic effects in bone tumors and metastasis[35]. Hence, we next sought to determine the role of DLX1 using an experimental bone metastasis model, wherein intramedullary tibial injection was performed using 22RV1-SCR control and *DLX1*-KO cells in athymic NU-*Foxn1*[nu] nude mice. Four weeks post-injection, X-ray scans were taken and subsequently, excised tibia implanted with tumor cells were subjected to micro-computed tomography (microCT), which showed higher bone loss in the control 22RV1-SCR group compared to *DLX1*-KO (Fig. 3f and Supplementary Fig 3j). Further, we examined the bone morphometric parameters of the metaphysis region of the tibia from both groups using CTAn (CT-Analyser) software. Interestingly, tibia implanted with *DLX1*-KO cells showed an increase in bone volume fraction (Bv/Tv), bone surface (BS), and trabecular number (TN) compared to tibia in the control group, thus signifying the presence of bone loss and destruction of bone architecture in 22RV1-SCR control group (Fig. 3g). Taken together, these findings provide a comprehensive understanding of DLX1-mediated oncogenicity and its possible role in PCa associated bone metastases.

**Elevated ERG and AR levels show positive association with DLX1 expression.** Overexpression of ETS transcription factor, ERG owing to *TMPRSS2–ERG* gene rearrangement is considered an early event in ~50% of PCa cases[1], thus we next examined the association of DLX1 with ERG using the publicly available TCGA-PRAD[36] cohort. Stratification of the clinical genomic data of PCa patients ($n = 498$) based on the expression of *ERG* revealed that most of the cases with higher *ERG* levels also exhibit increased expression of *DLX1* transcript (Fig. 4a). Next, using the UALCAN cancer OMICS database[37] we analyzed the overall survival probability of patients with varying expression of *DLX1* and its association with the well-known seven molecular PCa subtypes defined by TCGA[36]. Notably, *TMPRSS2-ERG* fusion-positive patients with higher levels of *DLX1* experienced lower survival probability compared to fusion-positive patients with low/medium *DLX1* expression (Supplementary Fig. 4a), indicating the possible existence of oncogenic cooperativity between DLX1 and ERG. Furthermore, a significant positive correlation between *DLX1* and *ERG* expression was observed in both MSKCC[38] and TCGA-PRAD cohorts (Supplementary Fig. 4b, c).

To further confirm these findings, we performed IHC, and RNA in situ hybridization (RNA-ISH) for ERG and *DLX1* expression, respectively using a tissue microarray (TMA) comprising 144 PCa patient specimens, and all but three of these patients were hormone naïve. The RNA-ISH staining patterns for *DLX1* were classified into four levels ranging from the score of 0–3, nearly ~60% of the patients were found positive for *DLX1* expression ranging from low to high (Fig. 4b, c). Specimens stained for ERG were stratified into ERG positive (ERG+) and negative (ERG-) categories (Supplementary Fig. 4d). In agreement with our in silico analysis, 44 out of 46 *TMPRSS2-ERG* fusion-positive cases (~96%) showed positive staining for *DLX1* expression (Fig. 4d, e and Supplementary Fig. 4e), wherein ~28% patients showed a high score for *DLX1* (*DLX1*[Hi], score 3), ~35% patients with moderate (*DLX1*[Me], score 2) and ~33% patients with lower *DLX1* expression (*DLX1*[Lo], score 1) (Fig. 4e), suggesting a possible role of ERG in *DLX1* regulation. While ~42% ERG-negative tumors also show *DLX1* expression, indicating the involvement of ERG-independent pathway in its regulation (Supplementary Fig. 4e). In terms of clinical staging, the percentage of tumors stained positive for both ERG and *DLX1* substantially increase from ~21% in low Gleason score (GS6) to ~33% in high Gleason score disease (GS9), similarly tumors stained positive only for *DLX1* expression (ERG–/DLX1+) also exhibit higher Gleason score (Fig. 4f). Thus, suggesting that the majority of the PCa patients harboring higher *DLX1* with/without ERG expression (ERG+/DLX1+ and ERG-/DLX1+) are associated with advanced-stage disease.

Next, we explored the status of AR expression in these patient samples and examined its correlation with DLX1 expression. To achieve this, we performed IHC staining for AR on the same TMA and stratified the specimens as AR-positive (AR+) and negative (AR–) based on the staining intensity (Supplementary Fig. 4f). Next, the specimens categorized as AR+ were further examined for the presence or absence of ERG expression and/or *DLX1* expression by RNA-ISH (Fig. 4g, h). Interestingly, we found that ~95% of the patients (42 out of 44) positive for AR and ERG (AR+/ERG+) showed *DLX1* expression, similarly ~50% of the AR+/ERG– patients (36 out of 72) were also positive for *DLX1* (Fig. 4i), while this percentage decreased (~23%) for the patients (6 out of 26) whose tumors were negative for both AR and ERG. However, only two patients which were AR-negative and ERG-positive showed *DLX1* expression (Fig. 4i). This data was re-examined by stratifying patients' specimens based on AR expression, and we found that ~67% of the AR+ patients also exhibit *DLX1* expression (Fig. 4j). While ~29% of the AR– patients show *DLX1* expression, indicating a possible role of some other regulatory factor(s) contributing to *DLX1* upregulation (Fig. 4j). Moreover, *DLX1* level alone gradually increased as a function of disease stage; ~37% positive in GS6 disease to ~67% positive in GS9 disease (Fig. 4k), similarly, ~51% in pT2c (pathologic tumor 2c) to 69% in pT3b stage (Fig. 4l).

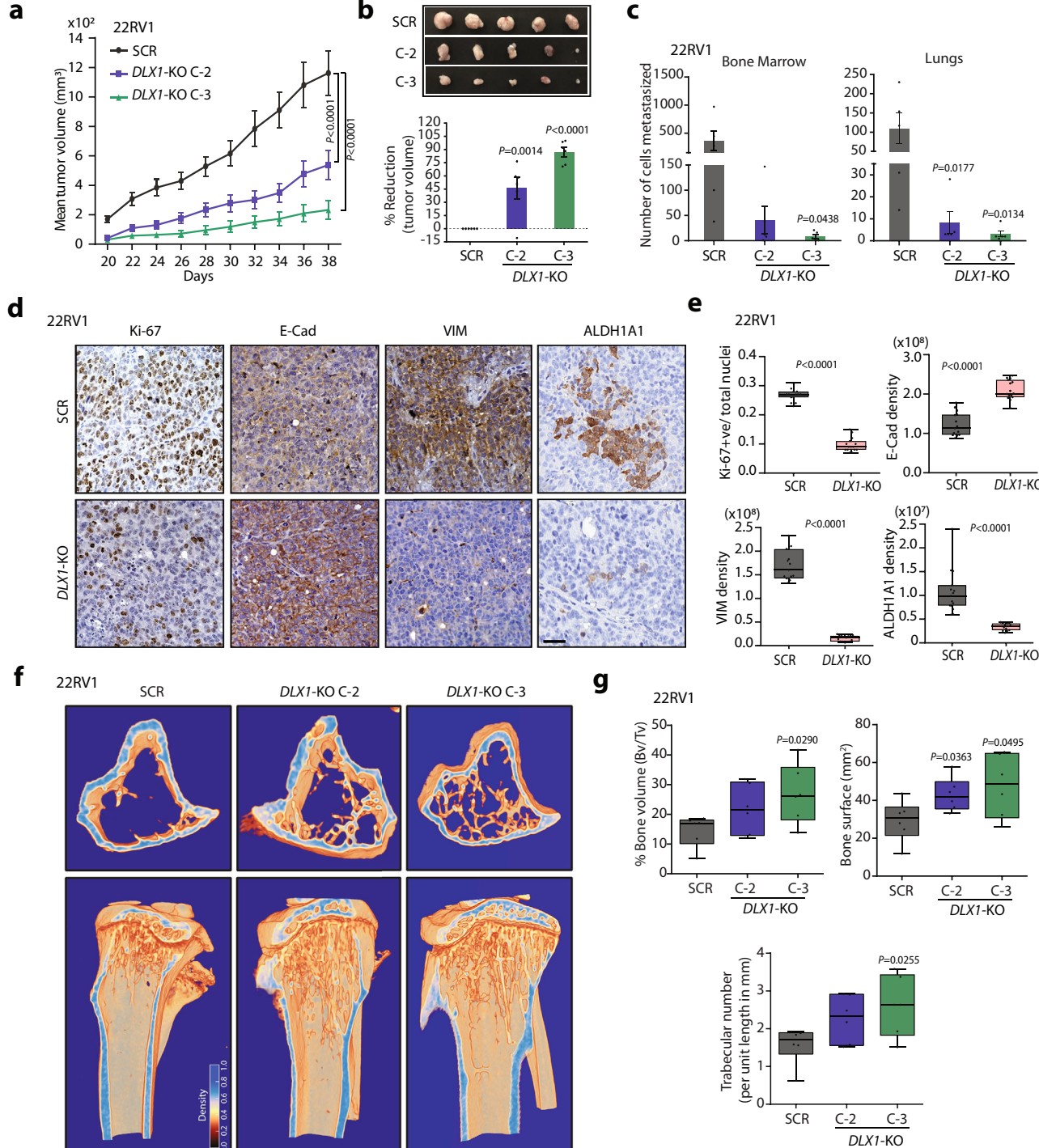

**Fig. 3 Abrogating DLX1 expression results in tumor regression and reduced metastases. a** Mean tumor volume of 22RV1-*DLX1*-KO and control SCR cells subcutaneously implanted in NOD/SCID mice ($n = 6$, $P < 0.0001$). **b** Representative images of the tumors excised at end of the xenograft experiment (top panel). Bar graph showing relative percent reduction in tumor burden. **c** Bar graphs representing number of cells metastasized to the bone marrow and lungs in xenografted mice as labeled ($n = 5$). **d** Images depicting immunostaining for Ki-67, E-cadherin (E-Cad), Vimentin (VIM), and ALDH1A1 on xenograft tumor sections, images are representative of three tissue samples. Scale bar, 35 μm. **e** Box plots showing immunostaining quantification of Ki-67, E-Cad, VIM, and ALDH1A1. Quantification was blindly done from 15 random histological fields ($P < 0.0001$). **f** Representative microCT bone images showing horizontal section (top panel) and vertical cross-section (bottom panel) views of the tibia excised from mice ($n = 6$) four weeks after intra-medullary tibia injection using 22RV1-*DLX1*-KO and control cells. **g** Same as **f**, except box plots showing bone architecture parameters analyzed using CTAn software. For panels **a**, **b** and **c**, data represent mean ± SEM. For panels, **a** Two-way ANOVA, Dunnett's multiple comparison test **b**, and **c** one-way ANOVA, Dunnett's multiple comparison test; **e** Unpaired two-tailed Welch's *t*-test; **g** Unpaired two-tailed Student *t*-test was applied. For panels, **e** and **g** Data are presented as box-and-whisker plots indicating median (middle line), 25th and 75th percentile (box) and minimum and maximum values (whiskers). Source data are provided as a Source Data file.

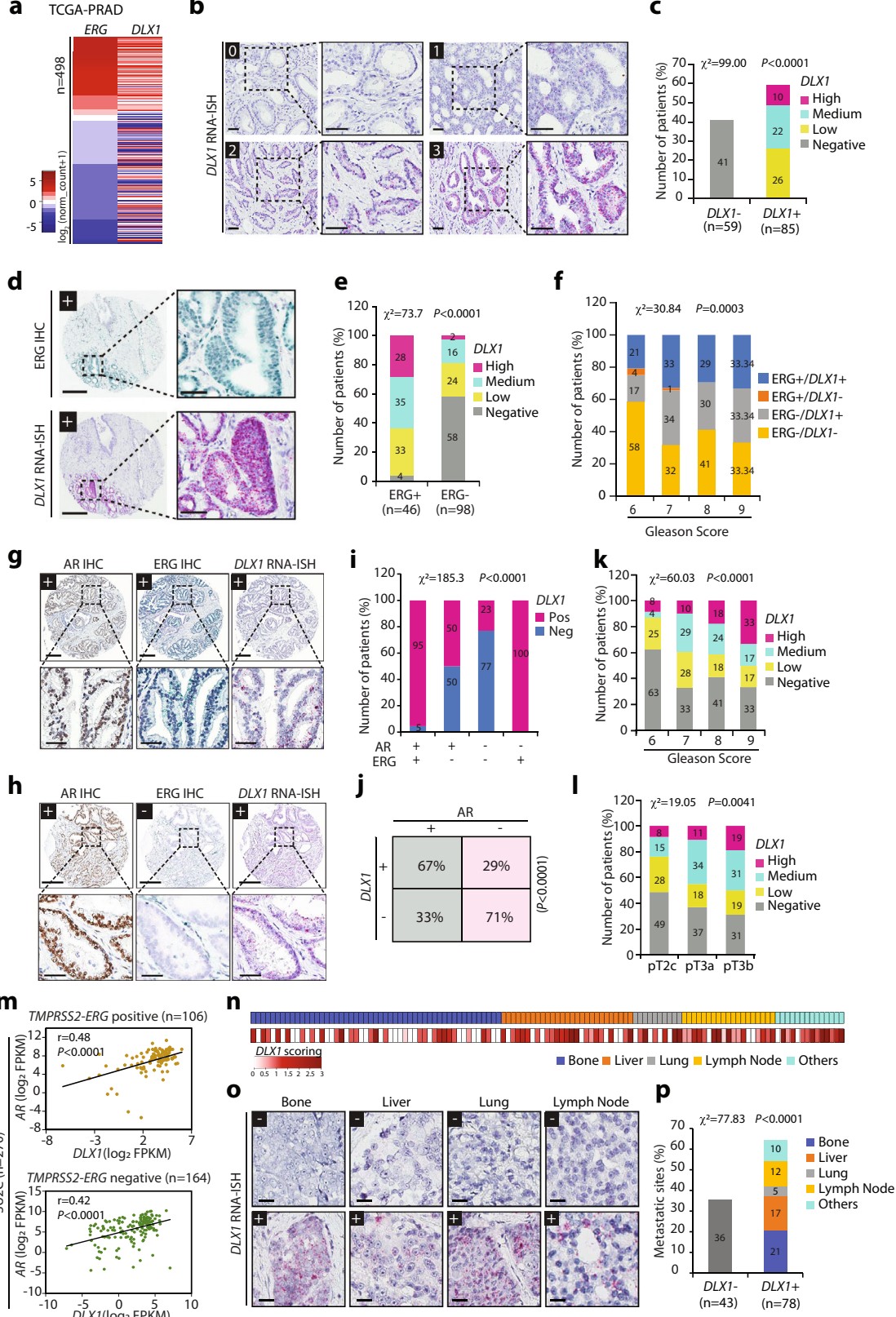

Providing the association of DLX1 with advanced-stage disease, we analyzed Stand Up to Cancer (SU2C) metastatic PCa dataset publicly available on cBioPortal for the expression of *DLX1* and its correlation with *AR* and *ERG*[39]. A significant positive correlation between *AR* and *DLX1* expression was observed in metastatic PCa patients irrespective of *TMPRSS2-ERG* fusion status (Fig. 4m).

Further, we performed RNA-ISH for *DLX1* in a metastatic PCa TMA comprising 121 metastatic sites collected from 45 patients (Fig. 4n, o). We found that ~64% of total metastatic sites showed *DLX1* expression, of these 21% belonged to bone, 17% to liver, 5% to lung, 12% to lymph node, and 10% to other organ sites (Fig. 4p). Together, our data suggest a role of DLX1 in aggressive and

**Fig. 4 Elevated ERG and AR correlates with higher DLX1 levels representing advanced-stage aggressive disease. a** Heatmap showing TCGA-PRAD RNA-Seq data for *ERG* and *DLX1* expression in primary PCa specimens ($n = 498$). Shades of red and blue represent the $\log_2$ (norm_count+1) value. **b** Representative core of PCa tissue microarray (TMA) showing RNA in situ hybridization (RNA-ISH) scoring pattern for *DLX1* in 144 PCa patient specimens, score 0 represents DLX1 negative, score 1 signifies low DLX1, score 2, and score 3 represents medium and high DLX1 expression, respectively. Scale bar, 50 μm. **c** Bar plot showing percentage of patients negative (DLX1–) and positive (DLX1+) for *DLX1* expression based on the scoring pattern ($P < 0.0001$). **d** Same as **b** except Immunohistochemistry (IHC) for ERG (top panel) and RNA-ISH for *DLX1* (bottom) in 144 PCa patient tissue specimens. **e** Bar plot showing percentage of patients with varying *DLX1* expression in ERG-positive (ERG+) and -negative (ERG–) PCa cases ($P < 0.0001$). **f** Same as **e** except an association between ERG and *DLX1* expression status and Gleason scores of PCa patients ($P = 0.0003$). **g** Same as **b** except representative tumor cores showing IHC for AR, ERG, and RNA-ISH for *DLX1* representing AR+/ERG+/DLX1+ status in 144 PCa patient tissue specimens. **h** Same as **g** except for representative AR + /ERG−/DLX1+ patient in TMA containing 144 PCa specimens. **i** Bar plot depicting percentage of patients with positive and negative *DLX1* expression in AR + /− and ERG + /− respective background. Patients showing low, medium, and high *DLX1* expression categorized as DLX1-positive ($P < 0.0001$). **j** Contingency table for the AR and *DLX1* status in TMA patient specimens. *P*-value denotes Fisher's exact test ($P < 0.0001$). **k** Bar plot showing association between *DLX1* expression and Gleason scores of tumor specimens ($P < 0.0001$). **l** Same as **k**, except association of *DLX1* expression with tumor stage ($P = 0.0041$). **m** Correlation plot of AR and *DLX1* using Stand Up To Cancer (SU2C) dataset by categorizing patients as *TMPRSS2-ERG* positive (top panel) and negative samples (bottom panel). *P*-value was calculated using two-tailed test with 95% confidence interval ($P < 0.0001$). **n** Heatmap showing *DLX1* levels in tumor specimens representing distant metastatic sites of metastatic CRPC patients. **o** Same as **n** except for RNA-ISH for *DLX1* expression in TMA containing 121 mCRPC biospecimens collected from various metastatic sites. Scale bar, 25 μm. **p** Bar plot showing DLX1 expression in percent metastatic sites from CRPC patients same as **n** ($P < 0.0001$). For panels **d**, **g**, and **h** scale bars are represented as 300 μm for the entire core and 50 μm for the inset image. For panels, **c, e, f, i, k, l,** and **p** *P*-value were calculated using Chi-Square test. Source data are provided as a Source Data file.

---

metastatic PCa and indicate a plausible oncogenic cooperativity between DLX1, ERG, and AR in the progression of this disease.

**ERG regulates *DLX1* in *TMPRSS2-ERG* fusion-positive prostate cancer.** Since we found a strong association between ERG and DLX1 expression in several independent PCa cohorts, we next sought to investigate the role of ERG in transcriptional regulation of *DLX1*. Thus, we analyzed publicly available ERG ChIP-Seq dataset in VCaP (*TMPRSS2-ERG* fusion-positive) cells[40,41], and an increased enrichment of ERG on the *DLX1* promoter was observed, while it was reduced in *ERG* depleted cells (Fig. 5a). Furthermore, enrichment of ERG on the *DLX1* promoter was also observed in ectopic ERG overexpressing RWPE1 cells[42] (Fig. 5a). Next, we scanned ~1 kb upstream and 500 bp downstream region to the transcription start site (TSS) of *DLX1* for the presence of probable ERG binding motif (EBM), and putative ERG binding sites namely, EBM1 (P1) and EBM2 (P2) were noticed (Fig. 5b). Next, our ChIP-qPCR data confirmed the significant recruitment of ERG at these EBMs on the *DLX1* promoter in VCaP cells (Fig. 5b). To confirm whether ERG occupancy is associated with transcriptionally active chromatin, we examined the presence of transcriptional activation marks, and a marked enrichment of H3K9Ac along with RNA polymerase II (RNA-Pol II) was observed (Fig. 5b). *PLAU*, an ERG target gene was used as a positive control. In agreement with this, ectopic ERG overexpression in RWPE1 cells (RWPE1-ERG) resulted in upregulation of DLX1 both at the transcript and protein levels (Fig. 5c). Furthermore, isogenic RWPE1 cells were examined for the ERG-mediated activation of the *DLX1* promoter by luciferase-based *DLX1* promoter reporter assay. As speculated, a significant increase in the reporter activity was observed in RWPE1-ERG cells transfected with a wild-type *DLX1* promoter reporter, while no significant change in the luciferase activity with mutated EBMs was observed (Fig. 5d).

Since AR signaling is known to drive the expression of ERG in *TMPRSS2-ERG* fusion-positive background, we next examined the effect of synthetic androgen methyltrienolone (R1881) on the expression of *DLX1* in VCaP cells. Notably, ~2-fold increase in DLX1 expression both at the transcript and protein levels was observed in R1881-stimulated VCaP cells (Fig. 5e, f). PSA was included as a positive control for androgen stimulation. Next, we tested whether AR also plays a direct role in the transcriptional regulation of *DLX1* (Fig. 5g). Thus, to ascertain this, we analyzed

ChIP-Seq datasets (GSE28951)[2] for AR and ERG in R1881-stimulated VCaP cells. Notably, a strong binding of AR on the third exon (chr2:172,661,000-172,662,500) of *DLX1* was observed in R1881-stimulated VCaP cells (Fig. 5h). On similar lines, several studies revealed the presence of exonic enhancers (eExons) which can act as a regulatory element for the nearby genes or the host gene on which they reside[43,44]. Hence, we propose that the presence of AREs at the third exon may act as a putative enhancer region in the transcriptional regulation of *DLX1*. We further examined the binding of AR on the *DLX1* gene using publicly available dataset (GSE70079)[45] comprising normal and PCa specimens, and a remarkable enrichment of AR was observed on the putative enhancer element of the *DLX1* in PCa specimens (Fig. 5i). Conclusively, we show that ERG directly gets recruited on the *DLX1* promoter thereby regulating its expression in *TMPRSS2-ERG* positive cases. Our findings also imply the potential role of AR signaling in mediating *DLX1* expression in PCa.

**AR regulates DLX1 expression in prostate cancer.** Considering an association between AR and DLX1 expression in PCa specimens and AR occupancy at the *DLX1* enhancer region, we next sought to examine the role of androgen signaling in the regulation of *DLX1*. Previously, presence of distinct AR binding sites (ARBS) were reported in tumor tissues than in normal prostatic tissue[45]. Furthermore, *DLX1* was found in the top ten upregulated genes in TCGA-PRAD dataset, which harbor tumor-specific ARBS in the neighboring 50 kb region (Supplementary Fig. 5a). To further validate the AR binding on the putative enhancer of *DLX1*, we performed ChIP-qPCR using R1881-stimulated VCaP cells, identifying a significant enrichment of AR on the *DLX1* enhancer, which was disrupted upon anti-androgen enzalutamide (Enza) treatment (Fig. 6a). The ARBS of *KLK3* was used as a positive control (Supplementary Fig. 5b). Further, we also examined the effect of another anti-androgen, EPI-001 on *DLX1* expression, a selective AR inhibitor that impedes transactivation of the amino-terminal domain (NTD) of AR thereby abrogating AR-V7 mediated transcriptional activity[46]. As speculated, treatment of VCaP cells with Enza and EPI-001 abrogated R1881-induced DLX1 expression, implicating the role of AR in transcriptional regulation of *DLX1* (Fig. 6b). Since FOXA1 is a known pioneer TF and a coactivator of AR[47], we next investigated the occupancy of FOXA1 at the *DLX1* putative enhancer region. Importantly,

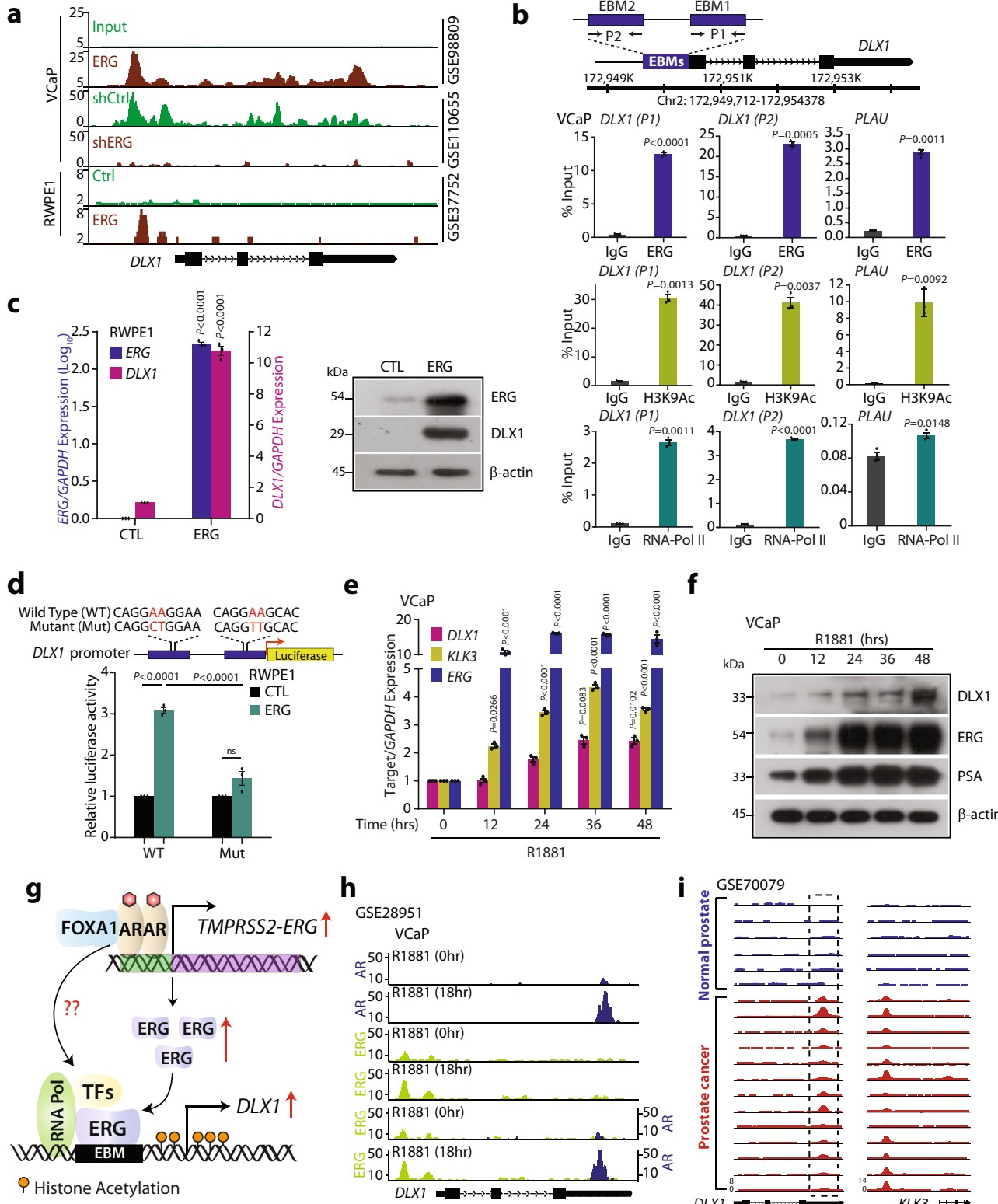

ChIP-qPCR in VCaP cells show FOXA1 enrichment on the same locus, signifying the recruitment of AR transcriptional complex on the putative enhancer region of *DLX1* (Fig. 6c). Similarly, ChIP-Seq analysis (GSE56086)[48] also revealed FOXA1 binding at the same locus (chr2:172,661,000–172,662,500) as occupied by AR (Supplementary Fig. 5c). Since AR and ERG interaction as a coregulatory TF is known to regulate the expression of common target genes[3,49], we examined the probable co-interaction of ERG

and AR at the *DLX1* promoter by performing re-ChIP experiment (Fig. 6d). As anticipated, ChIP using ERG antibody followed by pulldown using an AR antibody showed significant enrichment of AR and ERG at the *DLX1* promoter, thus confirming their interaction through the promoter-enhancer region (Fig. 6d). *CUTL2*, an AR-ERG co-regulated gene was used as a positive control. To further validate the plausible chromatin interaction at the *DLX1* genomic region, we analyzed the

**Fig. 5 DLX1 is a transcriptional target of ERG in TMPRSS2-ERG positive prostate cancer. a** Chromatin immunoprecipitation-sequencing (ChIP-Seq) data depicting ERG enrichment at the *DLX1* promoter in the indicated cell lines from GEO databases (GSE98809, GSE110655, and GSE37752). **b** Schema showing chromosomal location of the ERG binding motifs (EBM) onto *DLX1* promoter selected for ChIP-qPCR (top panel). ChIP-qPCR data showing recruitment of ERG, histone H3 lysine 9 acetylation (H3K9Ac), and RNA-polymerase II (RNA-Pol II) at the *DLX1* and *PLAU* promoters. **c** Q-PCR (left panel) and immunoblot (right panel) data showing the expression of ERG and DLX1 in RWPE1 cells overexpressing ERG (*P* < 0.0001). **d** Schema showing site-directed mutagenesis of *DLX1* promoter cloned upstream of luciferase gene, nucleotides in red were mutated (top). Luciferase reporter assay indicating wild-type (WT) and mutant (Mut) *DLX1* promoter-driven reporter activity (bottom panel) in ERG overexpressing and control RWPE1 cells (*P* < 0.0001). **e** Q-PCR data showing relative expression of target genes in VCaP cells stimulated with 10 nM R1881 at the indicated time points. **f** Same as **e**, except immunoblot data. **g** Schematic diagram depicting ERG-mediated transcriptional regulation of *DLX1* and plausible role of AR in DLX1 regulation. **h** ChIP-Seq data (GSE28951) showing recruitment of AR and ERG on the *DLX1* enhancer and promoter regions respectively, in R1881-stimulated VCaP cells. **i** ChIP-Seq data (GSE70079) showing enrichment of AR at the *DLX1* putative enhancer in the normal prostate (*n* = 6) and PCa (*n* = 13) tissue specimens. *KLK3* represents positive control. Data shown from three biologically independent samples (*n* = 3). Data represent mean ± SEM. For panels, **b** Unpaired two-tailed Welch's *t*-test; **c** and **d** Two-way ANOVA Sidak's multiple comparison test; **e** Two-way ANOVA, Dunnett's multiple comparison test was applied. Source data are provided as a Source Data file.

3D-chromatin landscape of RNA-Pol II in VCaP cells using ChIA-PET dataset (GSE121020)[50]. The integrative analysis of RNA-Pol II-associated peaks along with ChIP-Seq data of *DLX1* regulating TFs in PCa was performed. Consistent with our findings, the RNA-Pol II ChIA-PET data confirmed the promoter-enhancer interaction at the *DLX1* gene, which also indicated the binding of ERG, AR, and FOXA1 transcription factors (Fig. 6e). Owing to the promoter-enhancer interaction at *DLX1* gene loci and ERG occupancy at *DLX1* enhancer region in ChIP-Seq data (Fig. 6e), we carried out ChIP-qPCR for ERG at the *DLX1* enhancer region. As expected, we observed notable enrichment of ERG at the enhancer region of *DLX1* (Fig. 6f). Remarkably, siRNA-mediated knockdown of these regulatory factors namely, *ERG*, *AR* and *FOXA1* in VCaP cells (Supplementary Fig. 5d) resulted in reduced DLX1 expression across all siRNA conditions (Fig. 6g, h). Also, pronounced decrease in the level of DLX1 was achieved by concurrent silencing of all the three key regulators (*ERG*, *AR*, and *FOXA1*), thereby indicating the transcriptional interplay between these factors. Since the downregulation of FOXA1 is known to reprogram AR binding cistrome in PCa[45], we next examined if silencing *FOXA1* in VCaP cells redistributes binding of AR at *DLX1* enhancer region. Hence, we performed ChIP-qPCR for the AR occupancy in *FOXA1*-silenced VCaP cells and reduced AR enrichment at the *DLX1* enhancer region was observed, which was concordant with decrease in FOXA1 occupancy at these sites (Fig. 6i and Supplementary Fig. 5e). Collectively, our results suggest ERG and AR-mediated transcriptional co-regulation of *DLX1* in *TMPRSS2-ERG* fusion-positive cells. Furthermore, shRNA-mediated *AR* knockdown in castration-resistant and *TMPRSS2-ERG* fusion-negative, C4-2 cells also showed decrease in DLX1 expression (Supplementary Fig. 5f). Moreover, upon androgen stimulation, enrichment of AR and FOXA1 was observed in *TMPRSS2-ERG* fusion-negative 22RV1 cells, suggesting the significance of AR signaling in transcriptional regulation of *DLX1* in an ERG-independent manner (Fig. 6j).

Having established the higher DLX1 expression in CRPC cases and deciphering a substantial role of full-length AR in modulating its expression, we considered the possible involvement of AR-V7, an AR splice variant in transcriptional regulation of DLX1. Intriguingly, ChIP-Seq analysis for AR-V7 specific binding in 22RV1 cells revealed remarkable enrichment of AR-V7 at the putative enhancer region of *DLX1* (Supplementary Fig. 5g). Moreover, analysis of publicly available RNA-Seq data (GSE94013)[51] shows a significant reduction in *DLX1* transcript upon silencing AR-V7 or AR-FL in 22RV1 cells (Supplementary Fig. 5h) suggesting the role of AR-V7 in *DLX1* regulation. Furthermore, using an inducible expression system for AR-V7, we generated LNCaP cells that can express both

AR-FL and AR-V7, which mimics the clinical state of the majority of PCa patients with resistance to enzalutamide or abiraterone[52]. In these genetically engineered LNCaP cells, the expression levels of AR-FL and AR-V7 can be induced by treating them with R1881 and doxycycline (dox), respectively (Fig. 6k). In line with this, both AR-FL, as well as AR-V7 expression, led to increased expression of DLX1 transcript and protein levels (Fig. 6k, l). As we conjecture, treating LNCaP AR-V7 cells with AR antagonists (enzalutamide or EPI-001) in the presence of R1881 or dox abrogated AR-FL and AR-V7 mediated increase in DLX1 expression (Fig. 6m). Taken together, our findings suggest a role of AR-FL and AR-V7 in the transcriptional regulation of *DLX1* in ERG-dependent as well as -independent manner. These findings also highlight the critical role of AR-V7 in the transcriptional regulation of *DLX1*, thereby resonating with relatively high DLX1 expression in PCa patients with advanced-stage disease and higher Gleason score.

**BETi attenuates DLX1 expression and its oncogenic properties.** Having established ERG and AR signaling mediated transcriptional regulation of *DLX1*, we next explored the therapeutic strategies to target this regulatory circuitry. Since the utility of BETi, namely JQ1 and I-BET762 has been shown to inhibit aberrant AR signaling and the localization of BRD4 to AR target genes[12], we sought to investigate the ability of JQ1 in impeding the transcriptional regulators of *DLX1*, namely ERG and AR. Using ChIP-Seq dataset (GSE55064)[12], we examined the recruitment of ERG on *DLX1* promoter following BETi treatment in VCaP cells. Interestingly, a remarkable decrease in the enrichment of ERG was observed in JQ1 treated VCaP cells compared to control (Fig. 7a). Moreover, the presence of H3K27Ac marks indicates active *DLX1* promoter in the control VCaP cells, and *PLAU*, a known ERG target gene was used as a positive control (Fig. 7a). Further, these results were confirmed by ChIP-qPCR for ERG using JQ1 treated VCaP cells, and a similar trend was observed (Fig. 7b). We also looked for change in the DLX1 expression in JQ1 treated VCaP cells, and a significant decrease in its expression was observed (Supplementary Fig. 6a, b). To examine whether JQ1 can inhibit ERG-mediated DLX1 expression, stable RWPE1-ERG cells were treated with JQ1, and a reduced *DLX1* expression was observed, thereby confirming the efficacy of JQ1 in inhibiting ERG-mediated transcriptional regulation of *DLX1* (Supplementary Fig. 6c). Considering ERG-independent role of AR in *DLX1* regulation, we next treated C4-2, C4-2B, and 22RV1 cells with JQ1, and as speculated a significant reduction in the *DLX1* expression was noted (Supplementary Fig. 6d). Furthermore, VCaP and C4-2 cells treated with JQ1 showed a significant decrease in cell proliferation compared to vehicle control (Supplementary Fig. 6e). In addition, we

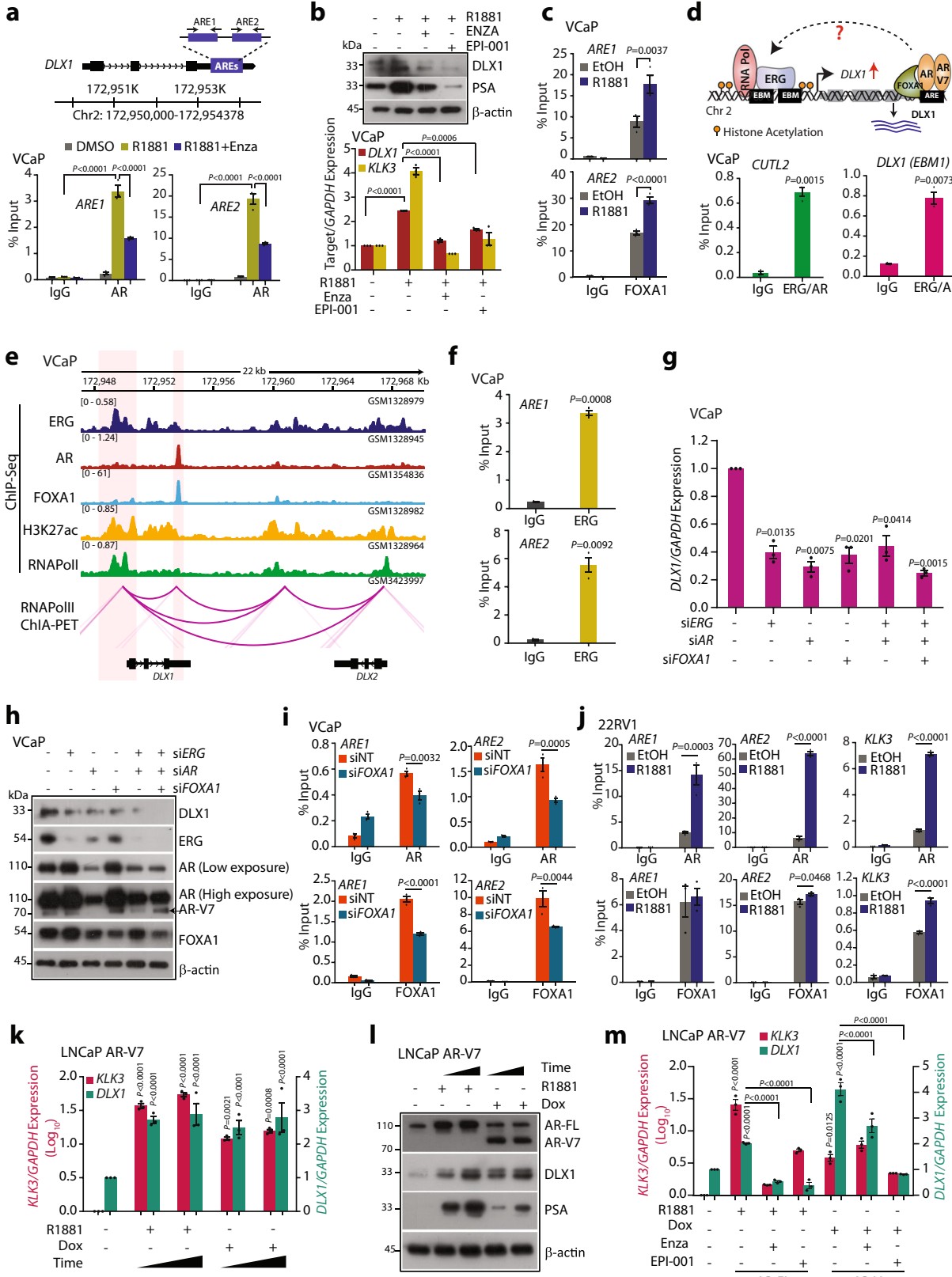

performed GSEA analysis of publicly available RNA-Seq data of JQ1 treated 22RV1 cells (GSE162564) using DLX1 putative target (Supplementary Fig. 3h) and observed enrichment of DLX1 target genes in control cells (Supplementary Fig. 6f). These data suggest downregulation of DLX1 putative target genes resulting in decreased oncogenic effect upon JQ1 treatment.

Recent preclinical and phase Ib/IIa clinical studies reported the efficacy of BET inhibitors in combination with anti-androgens in CRPC patients[13,53]. Hence, we examined whether JQ1 along with Enza could be effective in attenuating DLX1 expression and its oncogenic properties. Thus, we treated VCaP cells (*TMPRSS2-ERG* positive) with JQ1 alone or with

**Fig. 6 AR and AR-V7 regulate DLX1 expression in both ERG-dependent and -independent manners. a** Schematic showing the androgen response elements (AREs) at the *DLX1* putative enhancer (top panel). ChIP-qPCR data (bottom panel) depicting AR recruitment at the *DLX1* putative enhancer in R1881 (10 nM) stimulated VCaP cells in the presence or absence of Enzalutamide (Enza, 10 μM) ($P < 0.0001$). **b** Immunoblot (top panel) and Q-PCR (bottom panel) data showing relative expression of target genes in VCaP cells under similar culture conditions as indicated. **c** ChIP-qPCR data depicting FOXA1 recruitment at the *DLX1* putative enhancer in R1881 (10 nM) stimulated VCaP. **d** Schematic representation showing the possible interaction between ERG and AR on the *DLX1* promoter (top panel). Re-ChIP data showing co-enrichment of AR and ERG on EBM at the *DLX1* promoter (bottom panel). **e** Integrated genome view of 3D-chromatin structure and binding of transcription factors at the genomic and nearby region of *DLX1*. **f** Bar plots depicting ChIP-qPCR data for ERG occupancy at the *DLX1* enhancer region. **g** Q-PCR data showing relative expression of *DLX1* in siRNA-mediated *ERG, AR,* and/or *FOXA1*-silenced VCaP cells. **h** Same as **g** except immunoblot data. **i** ChIP-qPCR data showing enrichment of AR (top panel) and FOXA1 (bottom panel) in siRNA-mediated *FOXA1*-silenced VCaP cells**. j** ChIP-qPCR data depicting AR (top panel) and FOXA1 (bottom panel) enrichment at the *DLX1* putative enhancer in 22RV1 cells stimulated with R1881 (10 nM) for 16 h. *KLK3* shown as a positive control. **k** Relative expression of *KLK3* and *DLX1* in doxycycline (Dox) induced AR-V7 overexpressing LNCaP cells treated with R1881 (10 nM). For induction, 40 ng/ml of Dox or vehicle control was used for 24 and 48 h. **l** Immunoblots showing the expression of AR-FL, AR-V7, DLX1, and PSA using same cells as **k**. β-actin used as loading control. **m** Q-PCR data showing *DLX1* and *KLK3* expression in LNCaP AR-V7 cells under similar culture conditions as mentioned at 48-h time point. Data shown from three biologically independent samples ($n = 3$). Data represent mean ± SEM. For panels, **a, b** Two-way ANOVA Sidak's multiple comparison test; **f** Unpaired two-tailed Welch's *t*-test; **g** One-way ANOVA, Dunnett's multiple comparison test; **c, i, j, m** Two-way ANOVA, Tukey's multiple comparisons; **k** Two-way ANOVA, Dunnett's multiple comparison test was applied. Source data are provided as a Source Data file.

Enza and examined the expression of DLX1 and its target genes. Intriguingly, we observed ~60% reduction in *DLX1* expression as well as a concomitant decrease in the expression of DLX1 target genes, namely *ALDH1A1* and *HNF1A* in JQ1 treated VCaP cells compared to vehicle control (Fig. 7c). Although this effect was more pronounced (~90%) when VCaP cells were treated with JQ1 and Enza combination (Fig. 7c, d). Further, to investigate the JQ1 inhibitory effect in ERG-independent background, we treated castrate-resistant 22RV1 cells (ERG-negative) with both BETi and Enza. As speculated, a similar inhibitory effect in the expression of *DLX1, ALDH1A1,* and *HNF1A* was observed with JQ1 and/or Enza, indicating the absence of combinatorial additive effects as in case of VCaP cells (Fig. 7e, f). Nonetheless, PCa cells treated with Enza alone resulted in increased *HNF1A* expression which corroborates with the previous reports[26]. To investigate the efficacy of JQ1 in anti-androgen resistant cell line model, we performed similar experiments with enzalutamide-resistant 42D ENZ[R] cells (DLX1-positive and ERG-negative). As speculated, 42D ENZ[R] cells showed decrease in DLX1 expression in response to JQ1 treatment alone (Supplementary Fig. 6g, h). Further, we examined the effect of these drugs on the oncogenic properties in both ERG-dependent and -independent backgrounds, observing a marked reduction in cell proliferation rates of VCaP, 22RV1, and 42D ENZ[R] cells treated with JQ1 alone as anticipated. Combinatorial treatment with JQ1 and Enza exhibited a remarkable reduction in VCaP cell proliferation, but no such effect was noticed in 22RV1 and 42D ENZ[R] cells (Fig. 7g–i). Likewise, treatment of VCaP, 22RV1, and 42D ENZ[R] cells with JQ1 and/or Enza showed a decrease in the cell migratory and foci forming abilities, although this inhibitory effect was more evident in VCaP cells treated with the drug combination (Fig. 7j–l and Supplementary Fig. 6i–k).

Since we showed *ALDH1A1* as one of the DLX1 target genes, we next analyzed ALDH activity in response to JQ1 alone or combined with Enza in VCaP, 22RV1, and 42D ENZ[R] cells using flow cytometry-based assay. Interestingly, reduced levels of ALDH activity were noted in these cells treated with JQ1 alone, while drug combination showed slightly enhanced effect only in the ERG-positive VCaP cells (Fig. 7m–o). Since ALDHs play a critical role in the maintenance and differentiation of stem cells, we evaluated the efficacy of these two drugs in in vitro using a 3D-prostatosphere assay using VCaP cells. Notably, a significantly decreased number of spheres were formed in cells treated with drug combination than JQ1 alone (Fig. 7p). Moreover, reduced tumor-sphere forming ability of VCaP cells treated with drug combination group was corroborated by robust decrease in

*DLX1* expression compared to JQ1 alone (Fig. 7q). Interestingly, 22RV1 and 42D ENZ[R] cells failed to show enhanced effect in response to the combinatorial treatment with JQ1 and Enza, signifying the alternative pathways involved in *ERG* fusion-positive and -negative backgrounds. Collectively, our data reveal the efficacy of JQ1 alone or in combination with Enza for targeting DLX1-driven PCa in an ERG-dependent and -independent manner.

**BETi abrogates DLX1-mediated tumorigenesis and metastases in mice.** To investigate the efficacy of BETi against DLX1-mediated tumor growth in vivo, we implanted 22RV1 cells subcutaneously in athymic immunodeficient mice, and when the tumors reached a palpable stage (average volume ~75 mm³); mice were randomized into four groups ($n = 6$) and the drugs were administered. We observed that the mice treated with JQ1 alone or a combination of JQ1 and Enza showed almost similar trend in tumor regression (~50% and ~60% at day 19 and 25, respectively) as compared to the vehicle control group, indicating absence of any additive effect with the combinatorial treatment (Fig. 8a, b). Moreover, no adverse effect on the mice body weight was observed during the course of study (Fig. 8c). This observation is consistent with our in vitro data, wherein an additive anti-cancer effect with JQ1 and Enza combination was observed in VCaP cells, but not in ERG-negative 22RV1 cells. Furthermore, previous studies have also reported enhanced efficacy of JQ1 and Enza combination than JQ1 alone in VCaP tumors bearing mice[53]. Perhaps, these findings indicate that the additive anti-cancer effect of JQ1 and Enza combination is primarily augmented in *ERG*-positive background. Next, to determine the impact of drug treatment on spontaneous tumor metastasis, we excised lungs and bone marrow from the mice after terminating the study and isolated genomic DNA, followed by quantitative PCR for detecting human-specific *Alu*-sequence. We observed a marked decrease in the number of cells metastasized to these organs in both JQ1 alone, or in combination with Enza groups compared to vehicle control (Fig. 8d, e). Notably, no enhanced effect was observed in the group treated with the drug combination. Subsequently, to investigate the effect of these drugs in the tumor xenografts, we performed IHC staining for cell proliferative marker Ki-67, ALDH1A1, and RNA-ISH for *DLX1* expression. As speculated, a significant decrease in the *DLX1* expression, accompanied by reduced Ki-67 and ALDH1A1 expression was noted in JQ1 alone as well as in drug combination groups (Fig. 8f, g). Thus, our in vivo findings establish the therapeutic

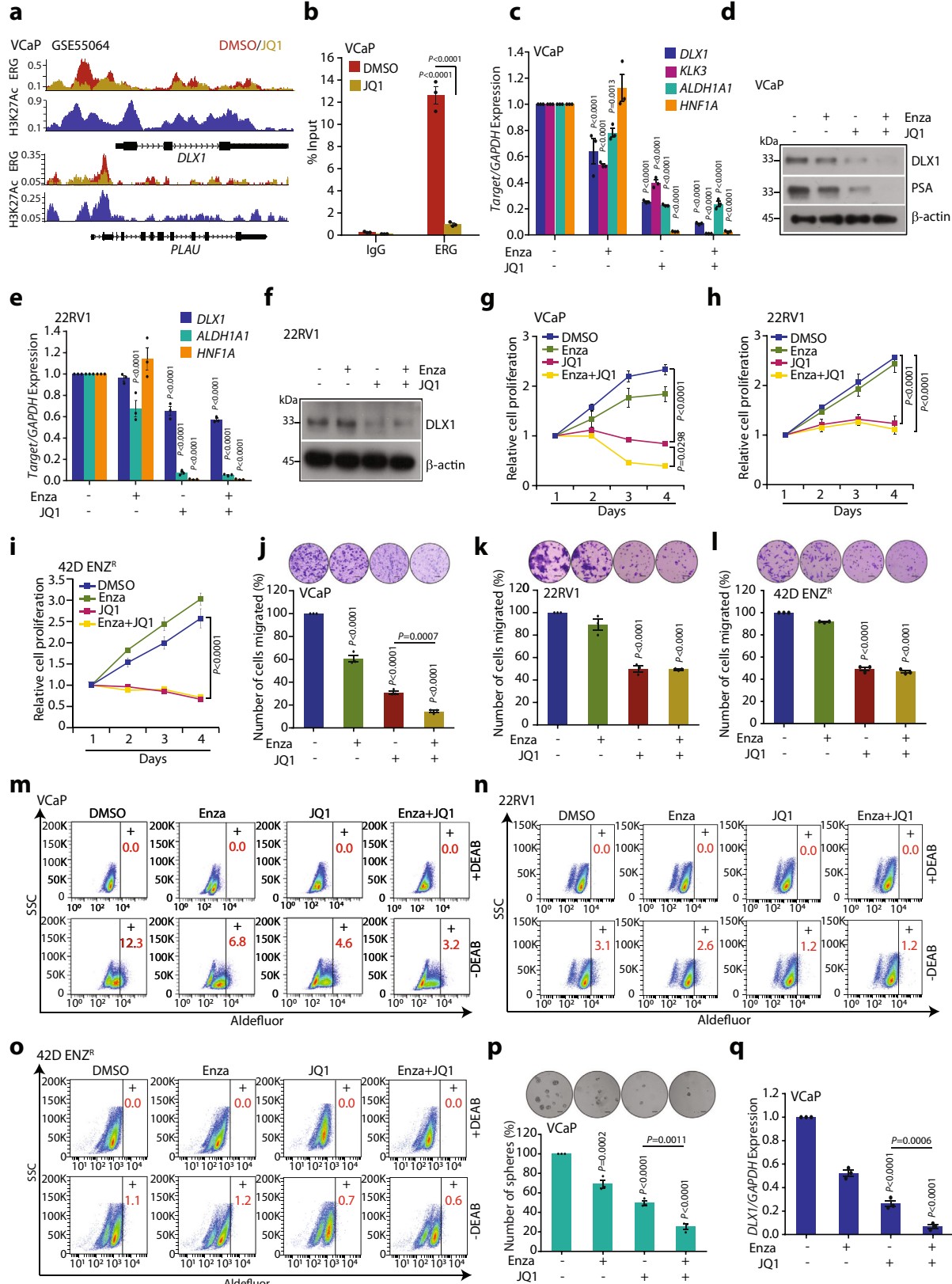

utility of BETi and anti-androgen in hampering the DLX1-mediated tumorigenesis and metastases.

Taken together, we unraveled ERG- and AR-mediated regulatory mechanisms involved in upregulation of DLX1 in an aggressive subset of PCa patients. We demonstrate that AR along with FOXA1 interacts with ERG as a coregulatory transcription factor, thereby orchestrating *DLX1* expression in *TMPRSS2-ERG* positive background. While, in fusion-negative background, AR and FOXA1 function in an ERG-independent manner, possibly in association with other coregulatory factor(s) to regulate the expression of *DLX1*. Thus, this triad of the key regulators control DLX1 expression in a context-dependent manner, resulting in

**Fig. 7 BET inhibitor alone or in combination with Enzalutamide downregulates DLX1 expression and mitigates its oncogenic properties. a** ChIP-Seq data (GSE55064) showing ERG enrichment on *DLX1* promoter in VCaP cells treated with JQ1 or vehicle control for 24 h. H3K27Ac represents active promoter in untreated cells. *PLAU* used as a positive control. **b** ChIP-qPCR data showing relative ERG enrichment on *DLX1* promoter in VCaP cells treated with JQ1 (0.5 µM) for 48 h ($P < 0.0001$). **c** Q-PCR data showing relative expression of target genes in VCaP cells treated with Enza (10 µM), JQ1 (0.5 µM) alone or in combination for 48 h. *KLK3* used as a positive control for JQ1 treatment. **d** Same as **c** except immunoblot. β-actin was used as loading control. **e** Same as **c** except 22RV1 cells. **f** Same as **e** except immunoblot ($P < 0.0001$). **g** Cell proliferation assay in VCaP cells treated with drug conditions as mentioned in **c**. **h** Same as **g** except 22RV1 cells ($P < 0.0001$). **i** Same as **g** except 42D ENZ^R cells ($P < 0.0001$). **j** Boyden Chamber Matrigel migration assay in VCaP cells using same treatment conditions as **c**. Inset shows representative image of the migrated cells (scale bar 30 µm). **k** Same as **j** except 22RV1 cells (scale bar 30 µm, $P < 0.0001$). **l** Same as **j** except Enza-resistant, 42D ENZ^R cells (scale bar 30 µm, $P < 0.0001$). **m** Fluorescence intensity of the catalyzed ALDH substrate in VCaP cells under same treatment conditions as **c**. Marked windows show ALDH1+ percent cell population. **n** Same as **m** except 22RV1 cells. **o** Same as **m** except 42D ENZ^R cells. **p** Bar plot showing number of spheres formed in prostatosphere assay using VCaP under same treatment conditions as **c**. Inset shows representative image of the spheres formed (scale bar 100 µm). **q** Q-PCR data for the *DLX1* expression using RNA isolated from VCaP prostatospheres. Data shown from three biologically independent samples ($n = 3$). Data represent mean ± SEM. For panels **b**, **c**, **e** two-way ANOVA, Tukey's multiple comparison test; **g**, **h**, **i** One-way ANOVA, Dunnett's multiple comparison test; **j–l**, **p**, **q** One-way ANOVA, Tukey's multiple comparison was applied. Source data are provided as a Source Data file.

DLX1-mediated oncogenesis via upregulation of several DLX1 target genes and biological processes. Importantly, we also demonstrate pharmacological inhibition of *DLX1* transcriptional circuitry by BET inhibitor in an ERG-independent background, and in combination with anti-androgen in an ERG-dependent context (Fig. 9).

## Discussion
Molecular biomarker-based diagnostic tests for DLX1 and HOXC6 in post-DRE urine samples have been instrumental in reducing unnecessary biopsies and identifying PCa patients at increased risk of high-grade disease[18,54]. Here, we show association of higher DLX1 expression in PCa patients' specimens with aggressive disease and overall poor survival. Integrative omics approaches such as Tracing Enhancer Networks using Epigenetic Traits (TENET) have identified DLX1 to be associated with over hundred tumor-specific active enhancers in primary PCa patients, thereby implicating the significance of the extensive DLX1 cistrome that contributes to tumor progression[23]. In addition, patient's positive for both ERG and DLX1 expression exhibit higher Gleason score and poor survival probability, emphasizing the oncogenic cooperativity which may contribute to disease aggressiveness and distant metastases. Similar to ERG-mediated transcriptional regulation of *DLX1*, TDRD1 (Tudor Domain Containing 1), another established PCa biomarker has been shown to be differentially regulated in *ERG* fusion-positive patients, wherein ERG modulates the methylation patterns of the *TDRD1* promoter thereby activating its transcription[55]. Furthermore, meta-analysis of gene expression data from five independent PCa studies indicated co-clustering of *ERG* with *DLX1* implicating the hierarchical gene regulatory network of transcription factors[56], thus suggesting that ERG might modulate the TFs involved in embryonic development. In corroboration with these independent studies, our findings draw attention to the hierarchical regulatory network between ERG and DLX1, wherein the emergence of *TMPRSS2-ERG* fusion as an early event in primary PCa results in direct transcriptional upregulation of DLX1 via ERG/AR contributing to disease progression. Collateral to this, we discovered an association between DLX1 and AR expression irrespective of *TMPRSS2-ERG* fusion status, and the role of AR signaling in transcriptional regulation of DLX1 in ERG-independent context.

Progression of CRPC is supported by sustained AR signaling which is predominantly dependent on the expression of constitutively active ligand-independent AR-V7[9]. Several studies reported AR transcriptional cistrome reprogramming in metastatic CRPC, wherein a cellular context-dependent transcriptional network operates downstream of AR which is distinct from the AR transcriptional program engaged in androgen-dependent stage[57]. Furthermore, genome-wide AR binding profiles[57] demonstrated a highly complex transcriptional circuitry, where AR possibly functions through enhancers distant from the promoters to impart regulation of target genes[2,3]. Although it has been shown that ERG represses AR expression and inhibits AR-mediated transcriptional regulation of canonical genes[3], several reports show the cooperativity between these TFs, where AR binds at the enhancer region and forms chromatin loop to interact with ERG at gene promoters thereby regulating the expression of downstream target genes[5,49]. Recent advancements in understanding the genomic interactions and chromatin looping using high throughput sequencing and conformation capture techniques provided important insight into the high-resolution 3D-chromatin landscape of normal and PCa cells[50]. Moreover, the crucial information about the mapping of RNA-Pol II, AR, and ERG in PCa demonstrates the long- and short-range interactions of these TFs orchestrating genomic expression in PCa[49,50]. In line with these studies, our data suggest the formation of an active transcriptional complex involving recruitment of AR-FL along with FOXA1 at the putative enhancer region of *DLX1*, which also interacts with ERG occupied *DLX1* promoter in fusion-positive cases, and possibly with other coregulatory TFs in the fusion-negative context. We speculate that this short-range chromatin loop structure between the enhancer-promoter region of *DLX1* is further enabled by the RNA-Pol II-associated interaction at *DLX1* genomic region, implicating the AR-driven transcriptional activation of *DLX1* in an ERG-dependent and -independent manner. Besides, our results also indicate the enrichment of AR-V7 at the putative enhancer region of *DLX1* and further reveal the role of constitutively activated AR-signaling in regulating its expression. Conclusively, these findings demonstrate the role of AR signaling wherein both AR-FL and AR-V7 regulate *DLX1* transcriptional activation in advanced stage PCa.

It is known that AR facilitates transcriptional regulation by partnering with transcriptional co-regulators such as histone deacetylases (HDACs), bromodomain-containing proteins (BRDs) as well as ETS TFs such as ERG[12,49,58]. Moreover, BETi are shown to attenuate AR and ERG-mediated oncogenesis in CRPC by disrupting transcriptional activation complex at their target gene loci[2,12,59]. Currently, therapeutic targeting of bromodomain proteins has gained clinical importance for the treatment of several malignancies including CRPC, for instance, BETi such as OTX-015, ZEN003694, and GS-5829 are already in clinical trials as single agents or in combination with anti-androgens for CRPC patients[60]. Alongside, our results demonstrate the potency of BETi alone and in combination with anti-androgen to reduce the expression of DLX1 and its target genes,

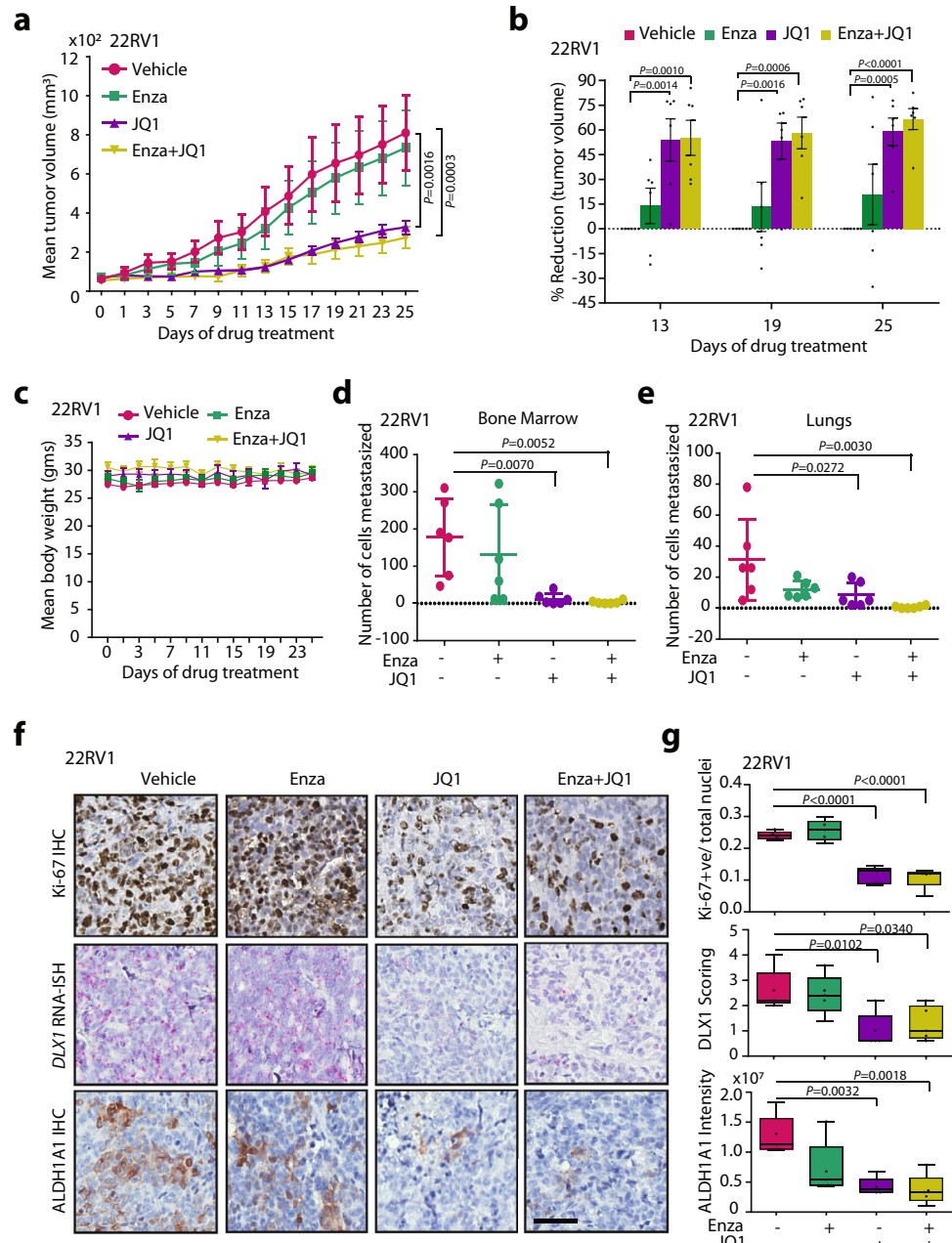

**Fig. 8 BET inhibitor alone or in combination with Enzalutamide attenuates DLX1-mediated tumorigenesis and metastases. a** Mean tumor volume of xenografts generated by implanting 22RV1 cells in athymic nude mice, and randomized into four treatment groups ($n = 6$ each), namely, vehicle control, Enza (20 mg/kg), JQ1 (50 mg/kg), and a combination of Enza and JQ1. **b** Bar plot showing percent tumor reduction in the treatment groups ($n = 6$) compared with the vehicle control group. **c** Mean body weight of mice ($n = 6$ per group) during treated with drugs as mentioned in **a**. **d** Scatter dot plot showing number of cells metastasized to the bone in xenografted mice treated with drugs ($n = 6$ per group) as mentioned in **a**. Data represent mean ± SD. **e** Same as **d** except cells metastasized to lungs ($n = 6$ per group). **f** Representative images depicting IHC staining for Ki-67, ALDH1A1, and RNA-ISH for *DLX1* using formalin-fixed paraffin-embedded tumor xenograft specimens ($n = 5$ per group) as **a**. Scale bar, 50 μm. **g** Box plots showing quantification of Ki-67, ALDH1A1, and *DLX1* expression in the tumor tissue sections ($n = 5$) of the mice xenografts as **a**. Data are presented as box-and-whisker plots indicating median (middle line), 25th and 75th percentile (box), and minimum and maximum values (whiskers). For panels **a** and **b** Data represent mean ± SEM. For panels, **a** two-way ANOVA, Tukey's multiple comparison test; **b** two-way ANOVA, Dunnett's multiple comparison test; **d**, **e**, **g** one-way ANOVA, Tukey's multiple comparisons test was applied. Source data are provided as a Source Data file.

thus, implicating the utility of DLX1 as a therapeutic target. Concomitantly, we also suggest that the enhanced effectiveness of drug combination (BETi and anti-androgen) is plausibly pre-dominant in *ERG* fusion-positive background. Our findings show that BETi along with anti-androgens could be used to mitigate the oncogenic effects of DLX1 via disrupting AR and ERG tran-scriptional circuitries and can be considered as a potential

therapeutic intervention in the treatment of the ERG+/DLX1+ subtype of PCa.

Although, *TMPRSS2-ERG* gene fusion is the highly prevalent genetic alteration in PCa, unlike gene fusions involving oncogenic kinases (e.g., RAF-kinase fusions)[61], transcription factors such as ERG are challenging to target. Recent advancements with the development of small molecular inhibitors, peptidomimetics[62],

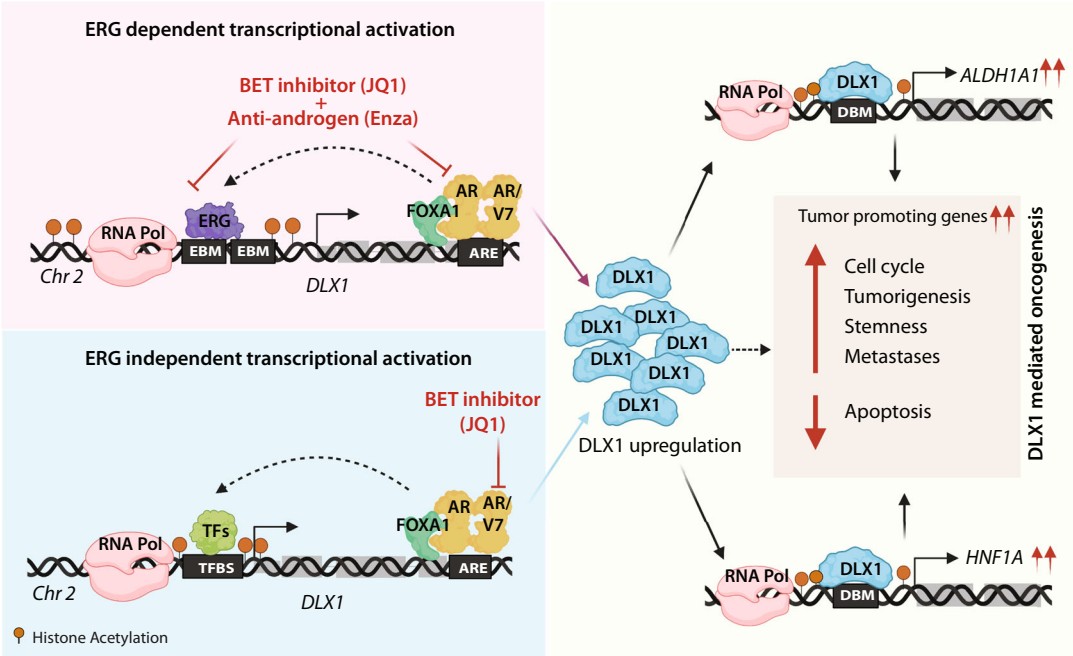

**Fig. 9 Schematic showing transcriptional regulation of *DLX1* and therapeutic utility of BET inhibitor and anti-androgen in suppressing DLX1-mediated oncogenic effects.** Illustration depicting transcriptional regulation of *DLX1* via ERG and AR-driven circuitry in *ERG* fusion-positive background, and through AR/FOXA1 in an ERG-independent context, thereby promoting cancer progression and metastases. In addition, DLX1 drives the expression of its target genes involved in stemness and oncogenesis, such as *ALDH1A1* and *HNF1A*. Utility of BET inhibitor alone or in combination with anti-androgen drugs to target transcriptional regulators of *DLX1* resulting in reduced expression of DLX1 and its downstream oncogenesis. Created with BioRender.com.

and proteolysis-targeting chimera (PROTAC)[63] show promise in targeting transcription factors in cancers, thus paving a way to exploit DLX1 as a potential drug target. Taken together, this study moved the field forward by providing strong evidence to employ parallel treatment regimens with BETi or/and AR targeted therapeutics for the clinical management of DLX1-positive PCa subtype, hence opening new treatment avenues for patients with advanced-stage disease.

## Methods

**In silico data processing and computational analysis**. To study the association of DLX1 with PCa, The Cancer Genome Atlas – Prostate Adenocarcinoma (TCGA-PRAD) dataset was downloaded from UCSC Xena browser (https://xenabrowser.net) and analyzed for *DLX1* gene expression. The samples were sorted according to the tissue type and data was plotted for *DLX1* expression ($\log_2$ (norm_count+1)) in solid normal tissue ($n = 52$) against primary PCa tissue ($n = 498$) using GraphPad prism version 7.0. No cut-off was applied on the dataset. Similar analysis was applied on the data retrieved from Gene Expression Omnibus (GEO) database with accession number GSE35988. For GSE80609, similar analysis was applied for TCGA-PRAD except FPKM values were plotted. For Kaplan–Meier survival analysis, survival data of patients with primary tumors were considered. Days to first biochemical recurrence and up to last follow-up for TCGA-PRAD patients were the parameters taken into account for the analysis. Samples were divided into two groups based on the expression level of *DLX1* using Cox proportional hazards regression model in R version 3.6.1. 5-year survival probability was calculated using Kaplan–Meier survival analysis by applying survival package (https://cran.r-project.org/web/packages/survival), and log-rank test was used to detect statistical significance. For survival analysis of patients with varying DLX1 expression and its association with other TCGA-defined PCa subtypes (including ERG), data was obtained from UALCAN database[37].

For the heatmap representation, hierarchical clustering was performed on expression data using heatmap.2 function of gplot package in R version 3.6.1. No cut-off was applied on the TCGA dataset for correlation plot of *ERG* and *DLX1*. Correlation plot between *ERG* and *DLX1* in TCGA-PRAD and MSKCC dataset was directly retrieved from GEPIA[64] and cBioPortal[65], respectively. SU2C data downloaded from cBioPortal was classified into two groups based on the *ERG*-fusion status, and the Pearson correlation graph was plotted using GraphPad prism.

To identify putative target genes of DLX1, publicly available ChIP-Seq data for DLX1 in LoVo colorectal cancer cell lines were analyzed, and the genes with enrichment of DLX1 were shortlisted and were overlapped with genes

downregulated in microarray expression data of 22RV1-*DLX1*-KO cells. Common genes present at the intersection were considered as DLX1 putative target genes.

**Bioinformatics analysis of sequencing data**. We analyzed the publicly available RNA-Seq data using Galaxy[66], a web platform available on the public server at https://usegalaxy.org. Raw sequencing FASTQ reads were prefetched and Fastq-dumped using SRA-toolkit (http://ncbi.github.io/sra-tools/). Followed by this, reads were assessed for their quality using FASTQC, followed by trimming of data using Trimmomatic. The adapter trimmed reads were next aligned to the human reference genome (hg19) using HISAT2 to obtain binary alignment map (BAM) files. Read alignment results were visualized using Integrative Genomic Browser (IGB)[67] in reference to the human genome. Next, the transcript abundance among different conditions was analyzed using FeatureCounts. Differential gene expression profiles were evaluated using DESeq2 to produce a list of differentially expressed genes with $\log_2$ fold change (FC) and FDR corrected *P*-values. Genes were annotated using the Entrez gene IDs and those with adjusted *P*-value < 0.05 were selected. Next, differential expressed genes were sorted such that genes with $\log_2$FC > 0 were considered upregulated while those with $\log_2$FC < 0 were considered downregulated. Pathway enrichment analysis was performed by using DAVID[68] and GSEA[69]. Volcano plot was generated using Galaxy online server.

ChIP-sequencing data analysis of online available datasets on GEO was performed. Galaxy web platform was used to perform ChIP-Seq analysis with default settings. Raw single-end reads in FASTQ format were uploaded to the web server using the NCBI SRA accession number for individual samples followed by FASTQC and sequence trimming with FASTQ Trimmer. Next, Sequence Alignment Map (SAM) files were generated by read alignment to the reference human genome (version as mentioned in the respective study) using Bowtie. Aligned reads were further filtered for the unmapped reads and were converted to the Binary Alignment Maps (BAM) file using SAMtools. ChIP-Seq peak calling was done using Model-based analysis of ChIP-Seq (MACS; $P < 10^{-5}$) data with default settings against respective controls. MACS output files were visualized using IGB. RNA-Pol II ChIA-PET data (GSM3423997) was downloaded from GEO and visualized using integrated genome viewer (IGV)[70].

**Cell lines culture conditions and authentication**. Prostate cancer cell lines (22RV1, VCaP, LNCaP, PC3) and benign prostate epithelial cells (RWPE1) were obtained from the American Type Culture Collection (ATCC) and were cultured as per the ATCC recommended guidelines. Using specific culture medium was supplemented with 10% fetal bovine serum (FBS) and 0.5% Penicillin-Streptomycin (Gibco Thermo-Fisher), cells were cultured in incubator supplied with 5% $CO_2$ at 37 °C (Thermo Scientific). Enzalutamide-resistant 42D ENZ[R] cells were obtained as a kind gift from Dr. Amina Zoubeidi[71].

For authenticating the cell line identity, short tandem repeat (STR) profiling was performed at the Lifecode Technologies Private Limited, Bangalore, and DNA Forensics Laboratory, New Delhi. *Mycoplasma* contamination test was routinely carried out for all the cell lines using PlasmoTest mycoplasma detection kit (InvivoGen).

**Establishing *DLX1* knockout cell line**. CRISPR/Cas9 knockout (KO) kit was purchased from Origene (KN206895) and KO cell lines were generated following the manufacturer's protocol. Briefly, pCas-guide vector containing guide RNA (gRNA) sequence (specific to *DLX1* and scramble control) along with donor vector containing homologous arms and functional cassette were co-transfected in the host 22RV1 cells. Cells were passaged up to eight generations post-transfection to eliminate the extrachromosomal donor vector and were then grown under puromycin (Sigma-Aldrich) selection (1 μg/ml) pressure to select the positive clones. Single-cell colonies were picked and cultured followed by screening at the genomic level. PCR was performed using the primers spanning the deleted genomic region in *DLX1* gene. The *DLX1-KO* cells were further validated for loss of DLX1 at the transcript and protein levels using qPCR and immunoblot, respectively (described elsewhere). Primer sequences are provided in Supplementary Table 1.

**Plasmids and constructs**. Briefly, DLX1 cDNA was cloned in pCDH lentiviral vector (Addgene) to establish DLX1 overexpressing cell lines. The pHAGE AR-V7 and control vectors were obtained as a kind gift from Dr. Nancy Weigel[72]. Successful lentiviral packaging of these constructs was performed to generate stable cell lines (described elsewhere). pGL3-Basic vector, a kind gift from Dr. Amitabha Bandyopadhyay at IIT Kanpur, was used to perform promoter reporter assay. For luciferase reporter assays, ~1 kb promoter region of *DLX1* was cloned in pGL3-Basic vector and site-directed mutagenesis was performed to mutate the ERG binding motifs (EBMs) in the *DLX1* promoter. Primer sequences used for mutagenesis are provided in Supplementary Table 1.

**Lentiviral packaging**. Lentiviral particles for pCDH vectors were generated using third-generation ViraPower Lentiviral Packaging Mix (Invitrogen) according to the manufacturer's protocol. For pHAGE vectors, second-generation pMD.2G and psPAX2 packaging systems were procured from Dr. Subba Rao Gangi Setty at the Indian Institute of Science, Bangalore. Briefly, packaging mix was co-transfected with lentiviral vector in HEK293FT cells using FuGENE HD Transfection Reagent (Promega). Media containing transfection reagent was replenished with complete growth media after 24 h. The lentiviral particles were harvested after 48–60 h, were aliquoted, and stored in −80 °C freezer.

For establishing stable cell lines, host cells were transduced with viral particles in a 6-well dish along with Polybrene (hexadimethrine bromide; 8 μg/ml) (Sigma-Aldrich) to increase the transduction efficiency. Media was replaced 24 h later, followed by selection of transduced cells under appropriate antibiotic pressure. To generate DLX1 overexpressing RWPE1 cells, pCDH-DLX1 vector was used for infection, and cells were selected in puromycin (0.5 μg/ml). LNCaP AR-V7 cells were selected under Geneticin (Gibco, ThermoFisher Scientific) at a concentration of 350 μg/ml. RWPE1-ERG overexpression and control cells were cultured in Keratinocyte SFM Media (ThermoFisher) along with supplements provided by the manufacturer.

**siRNA transfection**. VCaP cells plated at confluency of 40-45% in a six-well dish were transfected with 30 pmol of siRNA against *DLX1*, *ERG*, *AR*, *FOXA1*, and non-targeting control (Dharmacon). Lipofectamine RNAiMAX transfection reagent (ThermoFisher Scientific) was used according to the manufacturer's protocol. After 24 h, cells were transfected again, and 36 h post second transfection, they were harvested for functional assays. Likewise, transient *DLX1* knockdown was performed in 42D ENZ[R] cells. For ChIP-qPCR in VCaP, cells were plated in a 100 mm dish followed by transfection using siRNA against *FOXA1*.

**Functional assays**. For cell proliferation assay, about $1 \times 10^4$ cells per well were seeded in a 12-well culture dish, and cells were counted using hemocytometer at the indicated time points. Alternatively, Resazurin (Cayman Chemicals) was added to the cells ($2 \times 10^3$) plated in 96-well dish and fluorescence was measured with emission-excitation at 590–530 nm. The relative cell proliferation rate was plotted against the indicated time points.

Foci formation assay was performed by plating ~$2 \times 10^3$ cells in a six-well culture dish containing the recommended growth media in reduced serum (5% FBS) condition. Cells were incubated at 37 °C for three weeks by replacing media every third day. Paraformaldehyde (4% in 1× PBS) was used to fix the cells followed by staining with crystal violet solution (0.05% w/v), and number of foci were counted for quantification.

Cell migration assay was performed using 8 μm pore size of Transwell Boyden chamber (Corning). Growth media supplemented with 20% FBS was added to the lower compartment, followed by seeding $1 \times 10^5$ cells suspended in serum-free culture media to the Transwell inserts. Cells were incubated at 37 °C, 24 h later migrated cells were fixed in paraformaldehyde (4% in 1× PBS) and stained using crystal violet (0.5% w/v). Cells adhered onto Transwell filter were de-stained in 10% v/v acetic acid followed of quantification at absorbance at 550 nm. Migrated cells

were visualized and captured using upright Nikon microscope. For cells treated with drug, number of migrated cells stained with crystal violet were counted using ImageJ software.

For anchorage-independent soft agar assay, 0.6% low melting-point agarose (Sigma-Aldrich) dissolved in RPMI-1640 medium was poured in six-well culture dishes. After polymerization, another layer of 0.3% soft agar in RPMI-1640 medium containing cells (~$1.5 \times 10^4$) was poured on the top of the first layer. Cells were incubated for 20 days at 37 °C, and colonies >40 μm in size were counted.

**Real-time quantitative PCR**. To identify relative target gene expression, total RNA extracted using TRIzol (Ambion) was converted into cDNA using First Strand cDNA synthesis kit (Genetix Biotech Asia Pvt. Ltd) according to the manufacturer's instructions. Quantitative PCR (qPCR) was performed on the StepOne Real-Time PCR System (Applied Biosystems) using cDNA template, specific primer sets, and SYBR Green PCR Master-Mix (Genetix Biotech Asia Pvt. Ltd). Primer sequences are provided in Supplementary Table 1.

**Immunoblotting**. Whole-cell protein lysates were prepared in radio-immunoprecipitation assay buffer (RIPA buffer) supplemented with Phosphatase Inhibitor Cocktail Set-II (Calbiochem) and protease inhibitor (VWR), and protein sample was quantified by BCA assay, followed by protein separation on SDS-PAGE. Proteins were then transferred on the PVDF membrane (GE Healthcare), and the membrane was blocked with 5% non-fat dry milk in tris-buffered saline, 0.1% Tween 20 (TBS-T) for 1 h at room temperature. PVDF membrane was incubated overnight at 4 °C with the following primary antibodies at 1:1000 dilution: DLX1 (Thermo, PA5-28899), E-Cadherin (CST, 3195), Vimentin (Abcam, ab92547), phospho-Akt (CST, 13038), total-Akt (CST, 9272), Caspase-3 (CST, 9662), Cleaved PARP (CST, 9541), Bcl-xL (CST, 2764), ERG (Abcam, ab92513), FOXA1 (CST, 58613), 1:2000 diluted AR (CST, 5153), 1:2000 diluted PSA (CST, 5877), and 1:5000 diluted β-actin (Abcam, ab6276). Subsequently, PVDF membranes were washed with 0.1% 1× TBS-T followed by incubation at room temperature for 2 h with secondary anti-mouse or anti-rabbit antibody conjugated with horseradish peroxidase (HRP) (1:5000, Jackson ImmunoResearch Laboratories, Cat # 115-035-003 and Cat # 111-035-144, respectively). Membranes were again washed with 1× TBS-T buffer, and the signals were visualized by enhanced chemiluminescence system (Thermo) according to the manufacturer's instructions.

**Gene expression analysis**. Global gene expression profiling and identification of differentially regulated genes were performed as previously described[73]. Briefly, total RNA from 22RV1-*DLX1*-KO and 22RV1-SCR control cells was isolated and subjected to Agilent Whole Human Genome Oligo Microarray profiling (dual color) according to manufacturer's protocol using Agilent platform ($8 \times 60$ K format). Differentially regulated genes were filtered to only include significantly altered genes with ~1.6-fold average change ($\log_2$ FC >0.6 or <−0.6, $P < 0.05$). DAVID bioinformatics platform (https://david.ncifcrf.gov) was used to identify deregulated biological processes. GSEA was employed to detect gene enrichment in different datasets. Network-based analysis of biological pathways was generated using Enrichment Map, a plug-in for Cytoscape network visualization software[74](http://baderlab.org/Software/EnrichmentMap/).

**Flow cytometry**. For cell cycle distribution, cells were trypsinized and fixed in 70% ethanol followed by staining with propidium iodide (PI) (BioLegend, Cat # 421301) using manufacture's protocol. Cell cycle distribution was analyzed using in-built univariate model of FlowJo software. For apoptotic studies, cells were dissociated using StemPro™ Accutase™ (ThermoFisher Scientific) and were washed with cold 1× PBS and resuspended in 1× binding buffer ($1 \times 10^6$ cells/ml). Subsequently, $1 \times 10^5$ cells were stained using PE Annexin V Apoptosis Detection Kit I (BD Biosciences, Cat # 559763) following to the manufacture's protocol. Quadrants were gated on Annexin (PE) versus 7AAD (PerCP) channel dot plots using unstained, Annexin V (PE), and 7AAD (PerCP) single stained cells as controls. The quadrants were defined as lower left quadrant Annexin⁻/7AAD⁻ (viable), Annexin⁺/7AAD⁻ lower right quadrant (early apoptotic), Annexin⁻/7AAD⁺ upper left quadrant (necrotic), Annexin⁺7AAD⁺ upper right quadrant (late apoptotic) cells. Data were acquired on the Beckman Coulter's CytoFLEX platform and BD FACSMelody™ Cell Sorter and was analyzed using FlowJo software v10.6.1.

Aldefluor assay was performed to determine aldehyde dehydrogenase (ALDH) enzymatic activity using Aldefluor kit (Stem Cell Technologies, Catalog #01700) following manufacturer's guidelines. Briefly, cells were trypsinized and washed with 1× PBS followed by resuspension in 1 ml of Aldefluor assay buffer. Activated Aldefluor substrate (5 μL) was added to the cells and were divided in two conditions, with and without ALDH inhibitor, diethylaminobenzaldehyde (DEAB). After 30 min of incubation at 37 °C, cells were centrifuged and resuspended in 500 μL of Aldefluor assay buffer. The ALDH activity was detected in FITC channel. Gate was applied to identify viable-cell population using forward scatter (FSC) and side scatter (SSC) dot plot. Next, dot plot was generated with FITC channel versus SSC and gate on the control population was applied using DEAB treated samples. The same gate was applied over corresponding samples without DEAB to identify ALDH-positive population. For data acquisition, Beckman Coulter's CytoFLEX

platform and BD FACSMelody™ Cell Sorter were used. The analysis was performed using FlowJo software v10.6.1.

For stem cell markers, cells were stained with PE/Cy7 anti-human CD44 antibody (Miltenyi Biotec, 130-113-904, 1:50) and CD338-PE (ABCG2-PE, Miltenyi Biotec, 130-105-010, 1:40) followed by 1-h incubation at 4 °C. Firstly, gate was applied to identify viable-cell population using FSC and SSC dot plot. Histograms were generated for antibody-stained samples and were compared to their isotype controls. Data acquisition was carried out by using Beckman Coulter's CytoFLEX platform and BD FACSMelody™ Cell Sorter. Data were analyzed with FlowJo software v10.6.1. Gating strategies applied for the flow cytometry experiments are provided in Supplementary Fig. 7.

**Mice xenograft study**. NOD.CB17-$Prkdc^{scid}$/J (NOD/SCID) (Jackson Laboratory), five to six-week-old male mice were anesthetized using ketamine and xylazine (50 mg/kg and 5 mg/kg, respectively) cocktail injected intraperitoneally. 22RV1-$DLX1$-KO and 22RV1-SCR control cells were trypsinized and were suspended ($2 \times 10^6$) in 100 μl of saline with 20% Matrigel and implanted subcutaneously at the dorsal both flank sides of mice ($n = 6$ for each condition). Tumor burden was measured using digital Vernier caliper on alternate days, and tumor volume was calculated using formula ($\pi/6$) ($L \times W^2$), ($L$ = length; $W$ = width).

To examine spontaneous metastases from the xenograft study, lungs and bone marrow of the xenografted mice were excised at the end of the study and analyzed by performing Taqman assay using primers specific for human $Alu$-sequences as mentioned in Supplementary Table 1. Briefly, genomic DNA was isolated from the harvested organs and TaqMan probe FAM-YB8-ALU-167 (Applied Biosystems) with sequence 5′-6-FAM-AGCTACTCGGGAGGCTGAGGCAGGA-TAMRA-3′ (position 167–192) was used[31], which specifically hybridize to the human-specific YB8-Alu sequence. For generating standard curve Taqman assay was performed by using serially diluted human gDNA spiked with mouse gDNA, and accordingly based on the CT values number of metastasized cells were determined[31].

For experimental metastases experiment, athymic nude (NU(NCr)-$Foxn1^{nu}$), 5–6-week-old male mice procured from Hylasco Biotechnology Pvt. Ltd. (distributor for Charles River Research Models) were anesthetized. 22RV1-SCR or $DLX1$-KO cells ($4 \times 10^5$) were resuspended in 20 μl of saline and implanted by intramedullary tibial injections using 26-gauge needle. Piroxicam (3 mg/kg) was administered intramuscular post-implantation to reduce the pain. After 4 weeks, mice were subjected to X-ray scan and were euthanized. Tibia subjected to injections were harvested and analyzed using micro-CT.

To investigate the anti-tumorigenic effect of BETi and anti-androgen in mice xenograft model, 5–6-week-old male athymic nude (NU(NCr)-$Foxn1^{nu}$) mice were subcutaneously implanted with 22RV1 ($3 \times 10^6$) cells at both sides of dorsal flank. Mice were randomized into four groups ($n = 6$ per group) once the average tumor volume reached 75 mm³ and treated with vehicle control, enzalutamide (20 mg/kg), JQ1 (50 mg/kg) or JQ1 and Enza combination for 5 days a week. Enza diluted in 5% dimethyl sulfoxide (DMSO), 30% polyethylene glycol 400 (PEG-400) (Sigma-Aldrich), 65% corn oil (Sigma-Aldrich) was administered by oral gavage, while JQ1 diluted in 10% cyclodextrin (Sigma-Aldrich) was administered intraperitonially for 4 weeks. Tumor burden and mice weight was measured every alternate day. At the end of the experiment, mice were sacrificed after 4 h of drug treatment. Tumor, bone marrow, and lungs were excised for further characterization. Spontaneous metastasis to lungs and bone marrow was examined using TaqMan assay as described earlier. All immunodeficient mice used in the study were housed in specific-pathogen-free (SPF) facility as per the guidelines. The ambient temperature of the mice facility room was maintained at 20–24 °C with ~40–60% humidity and 12:12 h light–dark cycle.

**Micro-computed tomography (microCT)**. The analysis of the bone lesions and bone morphometric parameters was performed using microCT (Bruker microCT SKYSCAN). The scanning parameters used for performing microCT include resolution of 7 and 0.5 μm Al filter. Medium pixel setting with 48 kV and 204 μA of current with rotation step at 0.6 was used. The total time elapsed to scan each sample was 29 min. 3D image reconstruction and visualization were performed using volume rendering program CTVox. The CTAn v1.16.8.0 software was used for 3D image processing and parametric analysis.

**IHC of tumor xenografts and patients' specimens**. Tumor tissues excised from xenografted mice were fixed in 10% buffered formalin, followed by paraffin embedding and sectioning at 3 μm thickness using microtome (Leica). After deparaffinization and rehydration of tissue sections, heat-induced antigen-retrieval was performed using sodium citrate buffer (pH 6.0). Next, endogenous peroxidase activity was quenched using 3% hydrogen peroxide ($H_2O_2$) for 10 min, followed by blocking tissue sections in 5% normal goat serum, and were incubated with primary antibodies against Ki-67 (CST, 9449 S), E-Cadherin (CST, 3195), and Vimentin (Abcam, ab92547) at 4 °C overnight. After washing with tris-buffered Saline (TBS), sections were incubated with biotinylated secondary antibody at room temperature for 1 h followed by incubation with ABC (Avidin-Biotin complex) (Vector Labs) for 30 min. Sections were then processed for detection of HRP activity using DAB (3, 3-diaminobenzidine) peroxidase (HRP) substrate kit (Vector Labs). Quantification for Ki-67 using 15 random histological fields was performed

using ImageJS Ki-67 online module[75] and proliferation rate was calculated by normalizing Ki-67 positive nuclei with respect to total nuclei. For quantification of E-cadherin, Vimentin, and ALDH1A1 integrated density was calculated using IHC Toolbox in ImageJ software using 15 random histological fields.

Alternatively, PCa patient specimens and tumor xenograft slides were incubated at 60 °C for 2 h followed by target retrieval in a PT Link instrument (Agilent DAKO, PT200) using EnVision FLEX Target Retrieval Solution, High pH (Agilent DAKO, K800421-2). 1X EnVision FLEX Wash Buffer (Agilent DAKO, K800721-2) was used to wash slides followed by treatment with Peroxidazed 1 (Biocare Medical, PX968M) and Background Punisher (Biocare Medical, BP974L) for 5 min wash after each step. Slides were incubated with ERG [EPR3864] (Abcam, ab92513), AR (CST, 5153), Ki-67 (Agilent, IR626), and ALDH1A1 (CST, 54135) antibodies overnight at 4 °C. Afterward, slides were washed and then incubated in Mach2 Doublestain 1 (Biocare Medical, MRCT523L) for 30 min at room temperature, and then rinsed in 1X EnVision wash buffer and treated with a Ferangi Blue solution (Biocare Medical, FB813S). Next, slides were rinsed in distilled water followed by treatment with EnVision FLEX Hematoxylin (Agilent DAKO, K800821-2). After rinsing in tap water, slides were dried completely and then dehydrated using xylene. EcoMount (Biocare Medical, EM897L) was added to each slide, which was then mounted with coverslips.

**RNA in situ hybridization (RNA-ISH)**. For RNA-ISH, slides incubated at 60 °C were de-paraffinized in xylene. Slides were then kept in 100% ethanol twice for 3 min each and then air-dried following treatment with $H_2O_2$ for 10 minutes. Further, slides were rinsed and boiled in 1X Target Retrieval for 15 min. After rinsing slides in distilled water, Protease Plus treatment was given and then incubated with DLX1 probe (Advanced Cell Diagnostics, probe ID: 569601) for 2 h at 40 °C. Next, slides were washed and treated with Amp 1 for 30 minutes, Amp 2 for 15 min, Amp 3 for 30 min, and Amp 4 for 15 min, all steps were carried out at 40 °C in the HybEZ oven with two washes in 1× Wash Buffer. Slides were then treated with Amp 5 for 30 min and Amp 6 for 15 min at room temperature in a humidity chamber. Red color was developed by adding a 1:60 solution of Fast Red B: Fast Red A to each slide and incubating for 10 min. Finally, slides were treated with EnVision FLEX Hematoxylin (Agilent DAKO, K800821-2) and mounted using the same protocol as used for IHC slides.

**ERG and $DLX1$ staining evaluation**. ERG IHC staining was evaluated to define ERG positive and negative status of PCa tissues. $DLX1$ expression was identified by scoring the signal intensity of RNA-ISH probe hybridization for the TMA foci and the number of red dots/cell were evaluated to grade $DLX1$ expression into four levels ranging from score of 0–3 as described previously[76]. Next, an association between the expression of ERG and $DLX1$ was calculated by applying Chi-Square contingency test on GraphPad Prism version 7.0.

**Chromatin immunoprecipitation (ChIP)**. All ChIP experiments were performed as described previously[77]. Briefly, crosslinked cells were lysed using lysis buffer [1% SDS, 10 mM EDTA, 50 mM Tris-Cl, and protease inhibitor (Genetix)] followed by sonication for DNA fragmentation to an average fragment length of ~500 bp using Bioruptor (Diagenode). Sheared chromatin was incubated at 4 °C overnight with 4 μg of primary or isotype control antibodies. Antibodies against ERG (abcam, ab92513), H3K9Ac (CST, 9649), Rpb1 CTD/RNA PolII (CST, 2629), AR (CST, 5153), DLX1 (Thermo, PA5-28899), control rabbit IgG (Invitrogen, 10500C) and control mouse IgG (Invitrogen, 10400C) were used to perform ChIP assays. Simultaneously, the Protein G coated Dynabeads (Invitrogen) were blocked with BSA (HiMedia) and sheared salmon sperm DNA (Sigma-Aldrich), followed by incubation at 4 °C overnight. Blocked beads were incubated for 6–8 h at 4 °C with the lysate containing antibody to make antibody-bead conjugates. Next, the beads conjugated with antibody were washed and immunocomplex was eluted using elution buffer [1% SDS, 100 mM NaHCO₃, Proteinase K (Sigma-Aldrich) and RNase A (500 μg/ml each) (Sigma-Aldrich)]. DNA was isolated using phenol-chloroform-isoamyl alcohol extraction method. Binding sites for ERG transcription factor in the $DLX1$ promoter were identified using JASPAR and MatInspector software.

For Re-ChIP, first ChIP was performed using ERG antibody eluted in 10 mM DTT in Tris-EDTA buffer at 37 °C for 30 min, subsequently, DTT-elute was further diluted 50 times and second round of ChIP was performed following the similar protocol using antibody against AR. The ChIP-qPCR was performed using primers provided in Supplementary Table 1.

**Luciferase reporter assay**. Isogenic RWPE1-CTL and RWPE1-ERG cells plated in 24-well culture dish, and were transfected with pGL3-DLX1-P wildtype (250 ng) and pRL-null vector (2.5 ng) using FuGENE HD Transfection Reagent. After 48 h of incubation at 37 °C, cells were harvested using the lysis buffer provided with Dual-Glo Luciferase assay kit (Promega). GloMax® 96 Microplate Luminometer (Promega) was used to measure the Firefly and Renilla luciferase activity according to the manufacturer's protocol. Renilla luciferase activity was used as a normalization control. The same protocol was followed to measure the luciferase promoter reporter activity for pGL3-DLX1-P mutant construct.

**Androgen stimulation and drug treatment**. For androgen stimulation, cells were serum-starved for 72 h in phenol-red free media supplemented with 5% charcoal-stripped serum (CSS) (Gibco), followed by stimulation with 10 nM R1881 (Sigma-Aldrich) at the indicated time points. For anti-androgen treatment, VCaP cells were cultured in phenol-red free DMEM media supplemented with GlutaMAX (Gibco) and 5% CSS followed by pre-treatment with anti-androgen for 6 h. Next, cells were stimulated with 1 nM R1881 in the presence of anti-androgen for 48 h. The same procedure was followed for anti-androgen treatment in LNCaP cells, except RPMI-1640 phenol-red free medium (Gibco) was used. For ChIP-qPCR experiments, VCaP and 22RV1 cells were stimulated with 10 nM R1881 for 16 h. For VCaP anti-androgen ChIP experiment, cells were pre-treated with Enzalutamide for 6 h followed by stimulation with R1881 in the presence of anti-androgen for 16 h. For BETi and anti-androgen treatment, cells were grown in complete growth media followed by JQ1 and Enza treatment at 0.5 μM and 10 μM concentrations, respectively. The same concentration was used for combinatorial treatment. For functional assays and flow cytometry, cells were plated in six-well dish and treated with JQ1 and Enza at the above-mentioned concentrations for 48 h and were processed for further characterization.

**Prostatosphere assay**. VCaP cells ($2 \times 10^4$) were plated in low adherence 6-well dish and were cultured in serum-free DMEM-F12 media (1:1, Invitrogen) supplemented with B27 (1X, Invitrogen), EGF (20 ng/ml, Invitrogen), and FGF (20 ng/ml, Invitrogen). Cells were allowed to grow and the spheroids formed were collected and mechanically dissociated into a single-cell suspension before replating them in fresh media. The cells were then treated with DMSO, Enza, and/or JQ1 by adding them in the fresh media. Cells were passaged regularly and drugs were replenished in fresh media at each passage. Experiment was terminated after three weeks and the prostatospheres formed in all the groups were counted and imaged. At the end of the study, spheroids were harvested for isolating RNA as described earlier.

**Statistics**. For statistical analysis, one-way ANOVA, two-way ANOVA, unpaired two-tailed Student's t-test, and Unpaired two-tailed Welch's t-test was used for independent samples or otherwise mentioned in the respective figure legend. For P-value <0.05, the differences between the groups were considered significant. Significance was indicated as follows: *$P < 0.05$, **$P < 0.005$, and ***$P < 0.001$. Error bars represent the standard error of the mean (SEM) obtained from experiments performed at least three independent times.

**Study approval**. All mice experiment procedures were approved by the Committee for the Purpose of Control and Supervision of Experiments on Animals (CPCSEA) and abide to all regulatory standards of the Institutional Animal Ethics Committee of the Indian Institute of Technology Kanpur. TMAs comprising prostate cancer specimens ($n = 144$) were obtained from the Department of Pathology, Henry Ford Health System (Detroit, MI). These patients were not administered with any hormone therapy, except for three. Metastatic CRPC patients TMA was obtained from the Department of Urology, University of Washington, Seattle. Biospecimens were obtained within 8 h of death from patients who died of metastatic CRPC. Visceral metastases were identified at the gross level, bone biopsies were obtained from 16 to 20 different sites and metastases identified at a histological level. Tissues were fixed in buffered formalin (bone metastases were decalcified in 10% formic acid) and embedded in paraffin and were used to construct the TMA using duplicate 1 mm diameter cores from these tissues. Institutional Review Board approvals from the Henry Ford Health System (IRB#10375) and University of Washington (IRB#2341) and informed consents were received from the participants prior to inclusion in the study. All patients' specimens used in this study were collected in accordance with the Declaration of Helsinki.

**Reporting summary**. Further information on research design is available in the Nature Research Reporting Summary linked to this article.

## Data availability

The gene expression microarray data generated in this study has been submitted to the NCBI Gene Expression Omnibus (GEO), under the accession number GSE138738. There are various databases used in this study, namely: UCSC Xena (https://xenabrowser.net/) to download TCGA-PRAD dataset, cBioPortal (https://www.cbioportal.org/) to download MSKCC correlation plot and SU2C dataset, UALCAN cancer OMICS database (http://ualcan.path.uab.edu/) to generate survival plot of TCGA-defined PCa subtypes (including ERG) and DLX1, GEPIA (http://gepia.cancer-pku.cn/) to retrieve ERG and DLX1 correlation plot in TCGA-PRAD dataset. The source data of unprocessed gel images for Figs. 2b, c, 5c, f, 6b, h, l, 7d, f and Supplementary Figs. 1c, f, 3a, b, d, 5e, f, 6b, h is provided as Source Data file. Other datasets used in this study downloaded from the GEO (www.ncbi.nlm.nih.gov/geo) includes: Gene expression profiling in PCa patients: GSE35988; GSE80609; RNA-Seq data for RWPE1: GSE128399; RNA-Seq data for 22RV1 and VCaP cell: GSE118206; RNA-Seq data of DLX1 silenced C4-2B cells: GSE78913; ChIP-Seq data for ERG binding in VCaP cells: GSE98809; ChIP-Seq data for ERG binding in ERG silenced VCaP cells: GSE110655; ChIP-Seq data for ERG binding in RWPE1 ERG overexpressing cells: GSE37752; ChIP-Seq data for AR and ERG binding in

R1881-stimulated VCaP cells: GSE28951; ChIP-Seq for AR binding in PCa patient samples: GSE70079; ChIP-Seq in VCaP cells for ERG, AR, H3K27Ac, and RNA-PolII: GSE55062; ChIP-Seq in VCaP cells for FOXA1: GSE56086 ChIA-PET for RNA-PolII in VCaP cells: GSE121020; ChIP-Seq for ERG and H3K27Ac in JQ1 treated VCaP cells: GSE55064; ChIP-Seq for DLX1 in LoVo colorectal cancer cells: GSE49402; ChIP-Seq for FOXA1 and AR in R1881-stimulated VCaP cells: GSE56086; ChIP-Seq using AR-C19 and AR-V7 specific antibody in 22RV1 cells: GSE94013; RNA-Seq data of control, shAR-FL and shAR-V7 22RV1 cells: GSE94013; RNA-Seq data of JQ1 treated 22RV1 cells: GSE162564; Human reference genome (hg19) was downloaded from UCSC genome browser (https://hgdownload.soe.ucsc.edu/downloads.html#human). Source data are provided with this paper.

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

## Acknowledgements

This work is supported by the Science and Engineering Research Board (SERB), Government of India (EMR/2016/005273 to B.A.), and in part, by the DBT/Wellcome Trust India Alliance grant (IA/S/19/2/504659 to B.A.). B.A. acknowledges funding from the SERB-POWER grant (SPG/2021/000851 to B.A.) and S. Ramachandran-National Bioscience Award for Career Development (BT/HRD/NBA/NWB/39/2020-21 to B.A.) by the Department of Biotechnology. N.P. acknowledges financial support from CMDRP (W81XWH-16-1-0544). M.A. is supported by a Young Investigator Award from the Prostate Cancer Foundation (USA). We thank all the members of the Molecular Oncology group for their suggestions and critically reading the manuscript. We are thankful to Nancy Weigel for pHAGE AR-V7 construct, Paul Rennie for C4-2 shAR cells, Amina Zoubeidi for 42D ENZ$^R$ cells, Amitabha Bandyopadhyay for pGL3 basic vector, Jonaki Sen and Pradip Sinha for microscope facility, and Ashok Kumar for SKYSCAN microCT. We also thank Shraddha and Irfan Qayoom for their assistance with microCT. We thank the patients and their families who generously donated tumor tissues, Dr.'s Celestia Higano, Evan Yu, Pete Nelson, Heather Cheng, Elahe Mostaghel, Bruce Montgomery, Mike Schweizer, Lawrence True, Dan Lin, Eva Corey, and the rapid autopsy teams for their contributions to the University of Washington Prostate Cancer Donor Rapid Autopsy Program. The autopsy specimens are the result of work supported by resources by the Department of Defense Prostate Cancer Research Program (W81XWH-14-2-0183), the Pacific Northwest Prostate Cancer SPORE (P50CA97186), the PO1 NIH grant (PO1CA163227), and the Institute for Prostate Cancer Research. B.A.

is a Senior Fellow of the DBT/ Wellcome Trust India Alliance and a recipient of the S. Ramachandran-National Bioscience Award for Career Development.

## Author contributions

S.G. and B.A. conceptualized the study and designed the experiments. S.G. performed in vitro cell line-based studies. S.G. and V.B. performed the gene expression studies, bioinformatics analysis, and ChIP assays. S.G., V.B., and B.A. executed the in vivo mice xenograft studies. S.K. and T.B. performed in vitro assays and TaqMan assays for mice metastases experiments. S.G. and B.A. performed statistical analysis and interpreted the data. M.A. shared reagents for androgen signaling experiments. C.M. provided the metastatic CRPC patients' TMA. S.C., N.G., and N.P. provided the PCa TMA, performed immunohistochemistry and RNA in situ staining. S.G. and B.A. wrote the manuscript, which was reviewed by all authors. B.A. directed the overall project.

## Competing interests

The authors declare no competing interests.
