## [Peer Review File · Nature Communications]

Reviewers' Comments:

Reviewer #1:

Remarks to the Author:

In the present manuscript, Goel et al reported DLX1 as a gene that is up-regulated in prostate cancer and mCRPC as compared to normal and is associated with poor survival. They showed that DLX1 promotes oncogenic properties of PCa through regulating a variety of cellular processes including EMT, stemness, cell cycle, and apoptosis using in vitro and in vivo assays. Subsequently, the authors demonstrated DLX1 as a direct target gene of transcriptional regulation by ERG, AR, and FOXA1 through their binding on its promoter or enhancer. Lastly, they examined the ability of JQ1, which has previously shown to target ERG and AR activities, in suppressing DLX1. The manuscript is overall well presented. There are extensive analyses of previously published dataset to look at DLX1 expression and regulation in prostate cancer, which is very well done. While ERG regulation of DLX1 is novel, DLX1 has been previously reported as an androgen-regulated gene and a prostate cancer biomarker. The functional significance of DLX1 in promoting prostate tumor growth is novel and supported by the data. However, it did not move towards rigorous studies of DLX1 in promoting prostate cancer progression, metastasis, drug resistance, and its therapeutic targeting.

Major concerns:

1. Fig.2b: the increase of E-cad protein level does not seem to be in proportion with its mRNA level (Fig.2a).
2. Fig.2e-f: It is unclear how ALDH1A is related to DLX1. Fig.1-2 showed that DLX1 regulates a large number of genes and cellular processes. What's the underlying mechanism?
3. Fig.2m: does DLX1 regulate prostate metastasis considering its high expression in mCRPC and its regulation of EMT?
4. Fig.4i: it is interesting to see AR binding at DLX1 enhancer in prostate cancer. What's the status of ERG fusion in these samples and are there any correlation with AR enrichment?
5. Fig.5i-j: Dox did not induce DLX1 more than R1881 at mRNA level (Fig.5i), although it does increase protein level more than R1881 (Fig.5j) as the authors stated. This discrepancy should be addressed.
6. Fig.6g: JQ1 inhibiting prostate cancer growth is not new. On the other hand, does JQ1, alone or in combination with AR antagonists, abolish DLX1-induced tumor growth and metastasis in vitro and in vivo?
7. Why is DLX1 as an androgen-induced gene (Fig.4e) remains up-regulated in mCRPC (Fig.1b)?
8. How important is DLX1 in driving castration/enzalutamide-resistant PCa and in what percentage of mCRPC? Can DLX1-driven prostate cancer be targeted by JQ1 and/or enzalutamide?

Minor concerns:

Fig.1e/h: it will be useful to include the WB validating DLX1 overexpression or KO

Line 261: reference should be included for the dataset utilized.

Line 293: reference should be included for the dataset utilized.

Line 310/373: reference should be included for the dataset utilized.

Reviewer #2:

Remarks to the Author:

In this work, Goel et al. have examined the role of Distal-less-homeobox 1 (DLX1) in prostate cancer biology and the ways by which this is connected to the functions of other transcription factors such as ERG, androgen receptor (AR) and FOXA1. The paper contains an extensive amount of information from a plethora of experimental conditions. By and large, the authors' conclusions are supported by their own experimental data or those retrieved from the publicly available data resources. The experiments are carefully conducted and controlled, and the results appear to be reliable. Overall, the data presented in this paper are novel and potentially important for furthering our understanding of prostate cancer biology.

There are some relatively minor issues that the authors should address in order to strengthen this manuscript. These are listed below.

1. Line 86. In addition to bicalutamide and enzalutamide, the authors should list in this context

apalutamide and darolutamide and add appropriate references.

2. Line 262 onwards. Why did the authors examine in this context only DLX1 promoter for ERG binding, as in later experiments they identified a strong enhancer sequence within the DLX1 gene. Does ERG bind directly to this latter sequence?

3. The authors should cite, if possible, the original source articles for the GSE datasets that they refer to on multiple occasions in the manuscript. Thus, the inclusion of source articles as references would give the reader knowledge of the providers of the GSE datasets.

4. Line 356 onwards. The claim that "dox-inducible over expression of AR-V7 led to much higher expression of DLX1 than by R1881 stimulation alone" (Fig. 5i and 5j) may apply only to DLX1 protein accumulation (Fig. 5j) and not to DLX1 transcript accumulation (Fig. 5i).

5. In Fig. 5e, it is somewhat peculiar that silencing of ERG, AR and FOXA1 alone or in combination blunt the accumulation of DLX1 transcripts to the same degree. The authors need to comment on these results.

Since silencing of FOXA1 has been reported to elicit redistribution of AR-binding sites on PCa cell chromatin, the authors should examine whether or not silencing of FOXA1 modulates AR binding to the DLX1 enhancer.

Responses to Reviewers' comments

Reviewer #1 (Remarks to the Author):

In the present manuscript, Goel et al reported DLX1 as a gene that is up-regulated in prostate cancer and mCRPC as compared to normal and is associated with poor survival. They showed that DLX1 promotes oncogenic properties of PCa through regulating a variety of cellular processes including EMT, stemness, cell cycle, and apoptosis using *in vitro* and *in vivo* assays. Subsequently, the authors demonstrated DLX1 as a direct target gene of transcriptional regulation by ERG, AR, and FOXA1 through their binding on its promoter or enhancer. Lastly, they examined the ability of JQ1, which has previously shown to target ERG and AR activities, in suppressing DLX1. The manuscript is overall well presented. There are extensive analyses of previously published dataset to look at DLX1 expression and regulation in prostate cancer, which is very well done. While ERG regulation of DLX1 is novel, DLX1 has been previously reported as an androgen-regulated gene and a prostate cancer biomarker. The functional significance of DLX1 in promoting prostate tumor growth is novel and supported by the data. However, it did not move towards rigorous studies of DLX1 in promoting prostate cancer progression, metastasis, drug resistance, and its therapeutic targeting.

Response: We are thankful to the Reviewer #1 for his/her efforts in thorough evaluation of our manuscript and providing constructive suggestions. To address reviewer's concern about additional supporting data indicating the role of DLX1 in promoting prostate cancer progression, metastasis, drug resistance, and its therapeutic targeting, we have performed several new experiments in light of reviewer's valuable comments. New data addressing these points have been added to the revised manuscript, which has further strengthened this study.

Comment 1: Fig.2b: the increase of E-cad protein level does not seem to be in proportion with its mRNA level (Fig.2a).

Response: We understand reviewer's concern about E-cadherin protein level in the Figure 2b, we would like to clarify that this immunoblot was acquired at higher exposure time, thus the band for the E-cadherin protein got saturated, and doesn't seem conclusive (**Fig. 2b**). To address this issue, we repeated the immunoblot by loading lesser amount of freshly prepared protein lysate, and also reduced the exposure time of the x-ray film to avoid saturation of the protein band. In the revised manuscript, new immunoblot for the E-cadherin has been included (**Fig. 2b, in the revised manuscript**), which now reasonably looks proportionate with the mRNA level.

Comment 2: Fig.2e-f: It is unclear how ALDH1A is related to DLX1. Fig.1-2 showed that DLX1 regulates a large number of genes and cellular processes. What is the underlying mechanism?

Response: We appreciate reviewer's question regarding connection between ALDH1A1 and DLX1, and the mechanism contributing to deregulation of large number of genes and cellular processes in response to DLX1 ablation. In the revised manuscript, we have deciphered the connecting link between ALDH1A1 and DLX1, and also explored the molecular underpinnings involved in this process.

The Homeobox-containing genes such as *DLX1* are considered as master transcription factors, which regulates a wide range of downstream target genes involved in various aspects of morphogenesis, cell differentiation and developmental processes¹. For example, DLX1 is one

such important homeobox gene, which functions as a transcriptional regulator and play critical role in the development of mandible and ventral embryonic forebrain during embryonic development². Therefore, in the revised manuscript, we determined the DLX1 putative target genes in cancer background. By intersecting DLX1 downregulated genes in 22RV1-DLX1-KO cells and the DLX1-bound genes using ChIP-Seq data in LoVo colorectal cells (GSE49402)³, we generated a list of 677 common genes, which includes *ALDH1A1*, *HNF1A* and *GATA2* as putative target genes of DLX1 (**revised Supplementary Fig. 3h**). Additionally, DAVID analysis of DLX1 putative target genes showed pathways involved in cancer cell survival and progression (**revised Supplementary Fig. 3i**).

In the revised manuscript, we also demonstrated DLX1-mediated transcriptional regulation of *ALDH1A1*, which provides an explanation about reduced ALDH activity upon ablating DLX1 in prostate cancer cells. Additionally, we observed enrichment of DLX1 on the promoter regions of *ALDH1A1* and *HNF1A* gene, which was significantly reduced in *DLX1* knockout cells (**revised Fig. 2k-m**). Decrease in H3K9Ac, a transcriptional activation mark at same binding sites further confirms DLX1-mediated transcriptional regulation (**revised Fig. 2l-m**). Finally, reduced *ALDH1A1* expression was noticed in the tumor tissue excised from mice implanted with 22RV1-DLX1-KO compared to 22RV1-SCR control group (**revised Fig. 3d-e**).

Therefore, the new mechanistic data included in the revised manuscript inferred that DLX1 is an important transcription factor, which regulates several genes involved in prostate cancer progression.

Comment 3: Fig.2m: does DLX1 regulate prostate metastasis considering its high expression in mCRPC and its regulation of EMT?

Response: We thank the reviewer for asking this very important question. To address this, we have now added new *in vivo* data that identifies the role of DLX1 in regulating metastasis in prostate cancer. To examine metastasized human cancer cells in xenografted mice, TaqMan based quantitative PCR was performed to detect human specific *Alu*-sequence in DNA isolated from the bone marrow and lungs excised from the xenografted NOD/SCID mice. We found a significant reduction in the number of cells metastasized to the bone marrow and lungs in the mice implanted with 22RV1-DLX1-KO cells compared to control (**Fig. 3c in the revised manuscript**).

Moreover, DLX1 has been known to play crucial role in osteoclastogenesis and bone-resorption processes⁴, which contributes to osteolytic effects in bone tumor and metastasis⁵. Thus, to investigate the role of DLX1 in bone metastasis, we developed an experimental bone metastasis model in athymic nude mice by performing intramedullary tibial injection using 22RV1-DLX1-KO and control cells. We found that mice implanted with 22RV1-SCR control cells showed higher bone loss accompanied with decrease in bone volume fraction (Bv/Tv), bone surface (BS) and trabecular number (TN) compared to *DLX1*-KO group (**revised Fig. 3f-g**). This data confirms the role of DLX1 in distant metastases especially bone, which is considered as most preferential site of metastasis in PCa patients.

Comment 4. Fig.4i: it is interesting to see AR binding at DLX1 enhancer in prostate cancer. What is the status of ERG fusion in these samples and are there any correlation with AR enrichment?

Response: We appreciate reviewer's remark about *ERG*-fusion status of the patient samples used in the original **Fig. 4i** (**revised Fig. 5i**). This data was analysed by us using publicly available ChIP-Seq data (GSE70079)⁶, that showed AR enrichment at the *DLX1* putative

enhancer region, However, unfortunately the information about the *ERG*-fusion status for these patients is not available.

Nonetheless, in order to address reviewer's query, we have incorporated additional patient specimens' data showing the association of AR and DLX1 in ERG-positive and -negative context. In the revised manuscript, we performed IHC for AR in the primary PCa TMA, earlier it was immunostained for ERG and *DLX1* (**original Fig. 3**), and analysed the expression of ERG and *DLX1* in AR-positive and -negative PCa specimens (**revised Fig. 4g-i**). We found ~95% of the patients with AR and ERG positive expression exhibits *DLX1* expression, while ~50% of the AR+/ERG- patients show *DLX1* expression, indicating role of AR in regulation of *DLX1* in ERG-negative cases (**revised Fig. 4i**). Additionally, we have also incorporated RNA-Seq data of advanced stage metastatic PCa specimens obtained from the Stand Up to Cancer (SU2C) metastatic PCa cohort, wherein positive correlation between *AR* and *DLX1* expression was observed in *TMPRSS2-ERG* fusion-positive and -negative background (**revised Fig. 4m**).

Comment 5: Fig.5i-j: Dox did not induce DLX1 more than R1881 at mRNA level (Fig.5i), although it does increase protein level more than R1881 (Fig.5j) as the authors stated. This discrepancy should be addressed.

Response: We understand reviewer's concern and apologise for the inconvenience. To address this discrepancy, we have repeated this experiment multiple times, and we observed almost similar trend for DLX1 expression upon R1881 treatment or via DOX-induced AR-V7 overexpression at both mRNA and protein level. Therefore, in the revised manuscript, we have replaced the immunoblot data based on our most frequent observation, and have also removed the statement from the revised manuscript that "dox-inducible overexpression of AR-V7 led to much higher expression of DLX1 than by R1881 stimulation" (**Please refer to the revised Fig. 6k-l**).

Comment 6. Fig.6g: JQ1 inhibiting prostate cancer growth is not new. On the other hand, does JQ1, alone or in combination with AR antagonists, abolish DLX1-induced tumor growth and metastasis *in vitro* and *in vivo*?

Response: We thank the reviewer for asking this critical piece of data, as providing this information has further strengthen this study. In the revised manuscript, we have incorporated new data showing the effect of JQ1 alone or in combination with AR-antagonist enzalutamide (Enza) on DLX1 expression and its mediated oncogenesis by doing multiple sets of *in vitro* as well as mice xenograft experiments.

In the revised manuscript, we have added new data showing effect of JQ1 alone or in combination with Enza on the expression of DLX1 and its target genes in VCaP (ERG-positive), 22RV1 (ERG-negative) and 42D ENZ^R (ERG-negative and enzalutamide resistant) cell lines (**revised Fig. 7c-f and revised Supplementary Fig. 6g-h**). Also, we observed decrease in DLX1-mediated oncogenic properties upon JQ1 treatment alone or in combination with Enza in functional assays using these cell lines. Moreover, decrease in the ALDH activity was also observed in drug treatment conditions (**revised Fig. 7m-o**).

We have also performed xenograft study by subcutaneously implanting DLX1-positive 22RV1 cells in athymic nude mice, followed by administering drugs, namely JQ1 (50mg/kg), Enza (20mg/kg) and a combination of JQ1 and Enza. In line with our *in vitro* data, a significant reduction in tumor growth (**please refer to revised Fig. 8a-b**) accompanied with decrease in metastasis to bone marrow and lungs was noted (**revised Fig. 8d-e**). Additionally, immunostaining using tumor xenograft specimens showed decrease in DLX1 expression

followed by reduction in Ki67 and ADLH1A1 expression in JQ1 or/and Enza treatment groups (*revised Fig. 8f-g*).

Comment 7. Why is DLX1 as an androgen-induced gene (Fig.4e) remains up-regulated in mCRPC (Fig.1b)?

Response: There are several evidence in literature stating the significance of sustained AR signalling in metastatic CRPC stage⁷. Also, genomic profiling of the mCRPC patients revealed multiple aberrations in AR, such as gene amplification⁸, activating mutations⁹ and AR splice variants contributing to the persistent AR activity. The emergence of the AR splice variants including ligand independent AR-V7 results in constitutively active AR, which in the absence of androgen drives tumor progression¹⁰. Thus, these reports demonstrate the presence of active AR signalling and the regulation of AR downstream genes in mCRPC patients. On the similar lines, our data also show regulation of *DLX1* in response to AR signalling as well as via ligand independent variant AR-V7 (*please refer to revised Fig. 5e and Fig. 6*), hence rationalizing higher expression of DLX1 in mCRPC patients (*revised Fig. 1b and 4m-p*).

Comment 8. How important is DLX1 in driving castration/enzalutamide-resistant PCa and in what percentage of mCRPC? Can DLX1-driven prostate cancer be targeted by JQ1 and/or enzalutamide?

Response: We thank the reviewer for asking this pertinent question. To address this, we performed RNA-ISH for DLX1 expression in metastatic CRPC TMA comprising 121 metastatic sites specimens (*Revised Fig. 4n-p*). Interestingly, we observed that 64% (n=78) of the total metastatic sites showed DLX1 expression (*Revised Fig. 4p*). Additionally, we show that silencing *DLX1* expression in castrate-resistant PCa cells (22RV1 and VCaP) and in enzalutamide-resistant PCa cells (42D ENZ^R) result in reduced oncogenic properties and tumor growth (*Please refer to revised Fig. 2; Fig. 3 and revised Supplementary Fig. 3*). Together, these findings highlight the significance of DLX1 in driving advanced stage prostate cancer.

To further address reviewer's query, we have included functional assays data of DLX1-positive PCa cell lines, namely 22RV1, VCaP and 42D ENZ^R treated with JQ1 alone, and in combination with Enza. Our results reveal downregulation of DLX1 and its mediated oncogenicity in response to JQ1 or/and Enza in all the three tested cell lines. Decrease in the expression of DLX1 target genes was also observed upon drug treatment (*Please refer to revised Fig. 7, Fig. 8 and revised Supplementary Fig. 6*).

Minor concerns:

Fig. 1e/h: it will be useful to include the WB validating DLX1 overexpression or KO

Response: We apologise the reviewer for lack of clarity. Western blot data showing DLX1 overexpression and knockout is there in the *Supplementary Fig. 1c-f*, respectively.

Line 261: reference should be included for the dataset utilized.

Response: We have now included the references of the dataset used in this study.

Line 293: reference should be included for the dataset utilized.

Response: We have now included the references of the dataset.

Line 310/373: reference should be included for the dataset utilized.

Response: We sincerely apologise for the inconvenience. The references of the publicly available dataset are included in the revised manuscript.

Reviewer #2 (Remarks to the Author):

In this work, Goel et al. have examined the role of Distal-less-homeobox 1 (DLX1) in prostate cancer biology and the ways by which this is connected to the functions of other transcription factors such as ERG, androgen receptor (AR) and FOXA1. The paper contains an extensive amount of information from a plethora of experimental conditions. By and large, the authors' conclusions are supported by their own experimental data or those retrieved from the publicly available data resources. The experiments are carefully conducted and controlled, and the results appear to be reliable. Overall, the data presented in this paper are novel and potentially important for furthering our understanding of prostate cancer biology.

There are some relatively minor issues that the authors should address in order to strengthen this manuscript. These are listed below.

Response: We are thankful to reviewer #2 for his/her efforts in reviewing our manuscript, for giving overall positive assessment and constructive suggestions.

Comment 1. Line 86. In addition to bicalutamide and enzalutamide, the authors should list in this context apalutamide and darolutamide and add appropriate references.

Response: We thank the reviewer for pointing this. In the revised manuscript, we have now added the information about apalutamide and darolutamide along with appropriate references (Line 89-91).

Comment 2. Line 262 onwards. Why did the authors examine in this context only DLX1 promoter for ERG binding, as in later experiments they identified a strong enhancer sequence within the DLX1 gene? Does ERG bind directly to this latter sequence?

Response: We thank the reviewer for asking this important piece of data. In the revised manuscript, we have incorporated the new ChIP-PCR data depicting enrichment of ERG at the DLX1 enhancer region in VCaP cells (*revised Fig. 6f*).

Comment 3. The authors should cite, if possible, the original source articles for the GSE datasets that they refer to on multiple occasions in the manuscript. Thus, the inclusion of source articles as references would give the reader knowledge of the providers of the GSE datasets.

Response: We apologise to the reviewer for not providing this information. As suggested, we have included references of the original articles for all the GSE datasets used in this study.

Comment 4. Line 356 onwards. The claim that dox-inducible over expression of AR-V7 led to much higher expression of DLX1 than by R1881 stimulation alone; (Fig. 5i and 5j) may apply only to DLX1 protein accumulation (Fig. 5j) and not to DLX1 transcript accumulation (Fig. 5i).

Response: We thank the reviewer for his/her concern about discrepancy in the DLX1 expression at transcript and protein levels in response to R1881 treatment or via DOX-induced AR-V7. We have repeated this experiment multiple times, and almost similar trend for DLX1 expression upon R1881 treatment or via DOX induction was observed. Therefore, we have replaced the immunoblot data based on our most frequent observation, and have also removed the claim from the revised manuscript (*Please refer to the revised Fig. 6k-l*).

Comment 5. In Fig. 5e, it is somewhat peculiar that silencing of ERG, AR and FOXA1 alone or in combination blunt the accumulation of DLX1 transcripts to the same degree. The authors need to comment on these results. Since silencing of FOXA1 has been reported to elicit

redistribution of AR-binding sites on PCa cell chromatin, the authors should examine whether or not silencing of FOXA1 modulates AR binding to the DLX1 enhancer.

Response: We highly appreciate this remark by the reviewer on the original **Fig. 5e (revised Fig. 6g)**. Our findings revealed the role of ERG, AR and FOXA1 in transcriptional regulation *DLX1*. We showed that in *TMPRSS2-ERG* fusion-positive background, ERG along with enhancer-bound AR and FOXA1 forms the transcriptional activation complex and coordinates to drive the expression of *DLX1*. As rightly pointed by the reviewer#2 that siRNA-mediated silencing of *ERG*, *AR* and *FOXA1* alone or in combination blunts the accumulation of *DLX1* transcripts to the same degree, which indicates the plausible role of each regulatory factor in the activation complex. We speculate that silencing any of these factors perturb the regulatory complex, hence resulting in hampered expression of *DLX1* transcript to almost similar degree.

Moreover, considering the redistribution of AR-binding sites in PCa cells in response to FOXA1 silencing, we have performed ChIP-qPCR for AR in *FOXA1* silenced VCaP cells. In the revised manuscript, we have provided this data, wherein we found reduced occupancy of AR upon silencing *FOXA1* in VCaP cells (**revised Fig. 6i and revised Supplementary Fig. 5e**).

References:

1. Lindtner S, *et al.* Genomic resolution of DLX-orchestrated transcriptional circuits driving development of forebrain GABAergic neurons. *Cell reports* **28**, 2048-2063. e2048 (2019).
2. Petryniak MA, Potter GB, Rowitch DH, Rubenstein JL. Dlx1 and Dlx2 control neuronal versus oligodendroglial cell fate acquisition in the developing forebrain. *Neuron* **55**, 417-433 (2007).
3. Yan J, *et al.* Transcription factor binding in human cells occurs in dense clusters formed around cohesin anchor sites. *Cell* **154**, 801-813 (2013).
4. Lézot F, *et al.* Dlx homeobox gene family expression in osteoclasts. *Journal of cellular physiology* **223**, 779-787 (2010).
5. Wang M, Xia F, Wei Y, Wei X. Molecular mechanisms and clinical management of cancer bone metastasis. *Bone Research* **8**, 1-20 (2020).
6. Pomerantz MM, *et al.* The androgen receptor cistrome is extensively reprogrammed in human prostate tumorigenesis. *Nature genetics* **47**, 1346 (2015).
7. Coutinho I, Day TK, Tilley WD, Selth LA. Androgen receptor signaling in castration-resistant prostate cancer: a lesson in persistence. *Endocrine-related cancer* **23**, T179-T197 (2016).
8. Robinson D, *et al.* Integrative clinical genomics of advanced prostate cancer. *Cell* **161**, 1215-1228 (2015).
9. Grasso CS, *et al.* The mutational landscape of lethal castration-resistant prostate cancer. *Nature* **487**, 239-243 (2012).
10. Sharp A, *et al.* Androgen receptor splice variant-7 expression emerges with castration resistance in prostate cancer. *The Journal of clinical investigation* **129**, 192-208 (2019).

Reviewers' Comments:

Reviewer #1:

Remarks to the Author:

The authors have fully addressed all my concerns. The manuscript has been substantially improved with a number of additional experimental data.

Reviewer #2:

Remarks to the Author:

The Authors have responded to my main concerns in a satisfactorily fashion. I have no additional requests for revision.

Responses to Reviewers' comments

Reviewer #1 (Remarks to the Author):

The authors have fully addressed all my concerns. The manuscript has been substantially improved with a number of additional experimental data.

Response: We are thankful to Reviewer #1 for his/her positive assessment of our manuscript and providing constructive suggestions which helped in strengthening this study.

Reviewer #2 (Remarks to the Author):

The Authors have responded to my main concerns in a satisfactorily fashion. I have no additional requests for revision.

Response: We are thankful to the Reviewer #2 for his/her efforts in thorough evaluation of the manuscript and providing positive feedback on our work.